# Accelerating Discrete Langevin Samplers via Continuous Intermediates

## Abstract

Sampling from discrete distributions remains a challenge in machine learning, with traditional Markov Chain Monte Carlo (MCMC) methods such as Gibbs sampling suffering from inefficiency due to single-coordinate updates. Recent gradient-based discrete samplers have improved performance but remain constrained by the original discrete structures, which potentially hinder the convergence. To address this issue, we propose a hybrid approach that enables more global and informed proposals by introducing a continuous exploratory intermediate before the discrete update. This method, called Discrete Langevin Samplers via Continuous intermediates (cDLS), bridges the gap between discrete and continuous sampling and significantly accelerates convergence while maintaining theoretical guarantees. We develop variants of cDLS to ensure broad applicability, including unadjusted and Metropolis-adjusted versions. Experiments on Ising models, restricted Boltzmann machines, deep energy-based models, and Bayesian binary neural networks validate the superior performance of cDLS compared to existing methods. Our results highlight the potential of hybrid continuous-discrete exploration for advancing general discrete sampling.

## 1 Introduction

Discrete random variables are pervasive, from text data to complex genetic sequences. With the rapid progress of machine learning, the demand for effective sampling algorithms tailored to discrete distributions has become both urgent and essential. Markov Chain Monte Carlo (MCMC) remains a cornerstone for sampling from complex distributions, where its effectiveness highly depends on the proposal distribution. Among MCMC methods, Gibbs sampling is particularly popular for discrete distributions, yet its efficiency suffers in high-dimensional settings due to its coordinate-wise update scheme. To address this limitation, recent works have incorporated gradient information into discrete sampling, enabling more effective exploration and faster convergence (Grathwohl et al., 2021; Zhang et al., 2022; Sun et al., 2022b; 2023a;b; Xiang et al., 2023; Pynadath et al., 2024). These methods have demonstrated strong empirical performance and theoretical guarantees, advancing the state of the art in high-dimensional discrete sampling.

Although existing gradient-based discrete samplers benefit from the assumption of a differentiable extension, they do not fully exploit this smooth structure. Their updates remain restricted to discrete domains, which hinders larger moves and faster convergence, especially in high-dimensional spaces.

To fully harness smooth structure, we propose a hybrid method, *Discrete Langevin Samplers via Continuous Intermediates* (cDLS), which introduces a continuous intermediate to accelerate the discrete Langevin sampler (Zhang et al., 2022). By locally relaxing discrete variables into a continuous space, cDLS uses gradient information more effectively, enabling smoother and more directed transitions. Conceptually, it constructs auxiliary continuous paths to accelerate exploration: starting at a discrete point, traversing a continuous intermediate guided by gradient, and returning to the discrete space. A visualization of this process is shown in Figure 1.

By bridging discrete and continuous domains, cDLS inherits the efficiency of continuous Langevin dynamics while maintaining exact discrete updates. This hybrid design enables geometry-aware large moves and faster mixing, substantially improving convergence across various experiments. Notably, our framework is general

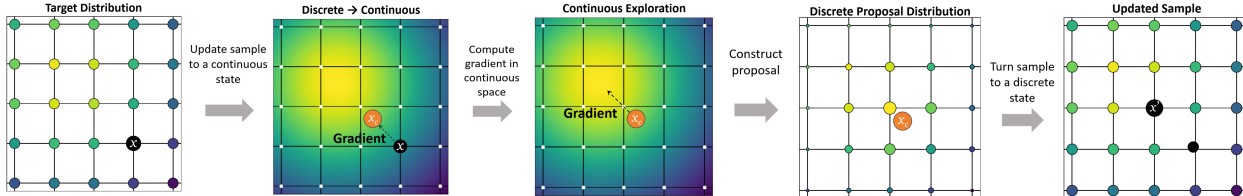

Figure 1: Overview of the proposed cDLS. Starting from a discrete state, the sampler performs a continuous update using gradient, then constructs a discrete proposal using discrete Langevin, and finally transitions to a new discrete state.

and can be integrated with a wide range of gradient-based discrete samplers. For clarity and brevity, we primarily focus our discussion on the Discrete Langevin Sampler[1], which serves as a representative example in the domain of discrete gradient-based sampling.

We summarize our contributions as follows:

- We propose discrete Langevin samplers via continuous intermediates (cDLS), a novel hybrid sampling framework that leverages continuous exploration to accelerate discrete Langevin sampling.

- We theoretically prove the correctness and efficiency of cDLS. We show that it achieves zero asymptotic bias for log-quadratic distributions without a Metropolis-Hastings correction, and it has small bias for distributions close to log-quadratic. Furthermore, we establish non-asymptotic convergence and inference guarantees for general discrete distributions.

- We demonstrate the effectiveness of our proposed algorithm on several experiments, including Ising models, restricted Boltzmann machines, deep energy-based models, and binary Bayesian neural networks. Notably, cDLS can achieve strong performance and low bias even without the use of Metropolis-Hastings correction.

## 2 Related Work

**Gradient-based Discrete Sampling** Efficient sampling from high-dimensional discrete distributions remains challenging due to the lack of informative proposals and poor scalability of traditional methods. Gradient-based approaches, such as Gibbs with Gradients (GWG)(Grathwohl et al., 2021) approximate locally balanced proposals (Zanella, 2020) using first-order Taylor expansion to guide coordinate selection, significantly reducing computational cost. Then, Sun et al. (2022b) introduced auxiliary path variables to extend local neighborhoods while maintaining gradient-based advantages. However, these methods still perform sequential updates and are limited by local neighborhood constraints. Inspired by Langevin dynamics in continuous space (Roberts & Stramer, 2002), Zhang et al. (2022) introduced a parallel update scheme for discrete sampling using a Langevin-like proposal. Follow-up works further extend this line by handling non-differentiable objectives (Xiang et al., 2023) and introducing adaptive scaling (Sun et al., 2023a; 2022a; Pynadath et al., 2024). In contrast, our work aims to accelerate gradient-based discrete sampling by more fully exploiting the underlying continuous structure, rather than relying solely on gradients. This introduces a new way to utilize continuous information, bridging the gap between discrete and continuous sampling approaches.

**Continuous Relaxation** Incorporating gradients in the proposal has been a great success in continuous space, such as the Langevin algorithm, Hamiltonian Monte Carlo (HMC), and their variants (Duane et al., 1987; Neal et al., 2011). To take advantage of this success, continuous relaxation is developed, which samples

---

[1]Our framework is general and can be readily applied to other gradient-based discrete samplers, such as Gibbs with Gradients (GWG) (Grathwohl et al., 2021). Due to space constraints, we present these extensions in Appendix D.

in a continuous space and then transforms the collected samples to the original discrete space (Han et al., 2020; Nishimura et al., 2020). As shown in previous work, this type of method usually does not scale to high-dimensional discrete distributions (Grathwohl et al., 2021). Relaxation-based methods conduct most sampling steps in the continuous space, with discretization applied only at the end. In contrast, our approach only incorporates a continuous intermediate step within each discrete update. This design preserves the discrete nature of the target space throughout the sampling process. The continuous intermediates in cDLS are not used to replace discrete sampling but instead act as local exploratory guides that improve discrete proposal construction then accelerate convergence.

## 3 Preliminaries

We aim to sample from a discrete target distribution of the form:

$$\pi(\theta) = \frac{1}{Z} \exp(U(\theta)), \quad \theta \in \Theta, \tag{1}$$

where $\theta$ is a $d$-dimensional variable, $\Theta$ is a finite discrete variable domain, $U$ is the energy function, and $Z$ is the normalizing constant for $\pi$ to be a distribution. In this paper, we have the following assumptions according to the literature on gradient-based discrete sampling (Grathwohl et al., 2021; Sun et al., 2022b; Zhang et al., 2022): (1) The energy function $U$ can be extended to a differentiable function in $\mathcal{R}^d$. (2) The domain is factorized coordinate-wise, $\Theta = \prod_{i=1}^{d} \theta_i$.

**Langevin Algorithm** The Langevin algorithm is a sampling method that uses gradient information with stochastic noise to generate samples from complex distributions. Given a target distribution 1, the Langevin dynamics are defined by the stochastic differential equation:

$$d\theta_t = \nabla U(\theta_t) dt + \sqrt{2} dW_t \tag{2}$$

where $W_t$ denotes the standard Brownian motion. The discretized version with stepsize $\alpha$ updates samples through:

$$\theta_{t+1} = \theta_t + \frac{\alpha}{2} \nabla U(\theta_t) + \sqrt{\alpha} \xi_t \tag{3}$$

where $\xi_t \sim \mathcal{N}(0, I_{d \times d})$. This algorithm efficiently explores probability spaces by combining deterministic gradient with Gaussian noise, making it widely applicable in Bayesian inference, energy-based models, and other machine learning tasks (Roberts & Stramer, 2002). Despite its simplicity, Langevin sampling often mixes slowly and is sensitive to hyperparameters, which limits its practical use compared with Hamiltonian Monte Carlo. These issues become more severe in discrete spaces where gradients are undefined. This gap motivates our proposed framework, which bridges discrete and continuous domains to retain the efficiency of Langevin dynamics while enabling exact discrete updates.

**Discrete Langevin Proposal** Discrete Langevin Proposal (DLP) is a counterpart of the Langevin algorithm in discrete domains (Zhang et al., 2022). Given a current state $\theta$, the proposal for $\theta' \in \Theta$ is

$$q(\theta'|\theta) = \frac{\exp\left(-\frac{1}{2\alpha} \left\| \theta' - \theta - \frac{\alpha}{2} \nabla U(\theta) \right\|_2^2\right)}{Z_\Theta(\theta)}, \tag{4}$$

where $Z_\Theta(\theta)$ is the normalizing constant over the discrete domain. It is worth noting that if $q(\theta'|\theta)$ were defined over $\mathbb{R}^d$ instead of $\Theta$, it would correspond exactly to a Gaussian distribution with mean $\theta + \frac{\alpha}{2} \nabla U(\theta)$ and covariance $\alpha I$, which is one of the reasons why this method is referred to as the discrete Langevin sampler.

The proposal admits a coordinate-wise factorization, allowing updates to be performed in parallel:

$$q(\theta_i'|\theta) \propto \exp\left(\frac{1}{2} \nabla U(\theta)_i (\theta_i' - \theta_i) - \frac{(\theta_i' - \theta_i)^2}{2\alpha}\right), \quad \theta_i' \in \Theta_i. \tag{5}$$

Like other gradient-based discrete sampling methods, DLP performs updates only in discrete space, which limits its ability to exploit the smooth structural information in the target distributions.

## 4 Discrete Langevin Sampler via Continuous Intermediates

In this section, we propose the *Discrete Langevin Samplers via Continuous Intermediates* (cDLS), a novel discrete sampler that improves discrete Langevin sampling by leveraging continuous intermediates. The core idea is to temporarily escape the discrete domain via a gradient-informed extension in continuous space, and then return to the discrete space through a projection and a discrete update. This allows the sampler to better utilize the underlying geometry of the energy function $U$ to facilitate more informed and exploratory updates. Unlike existing methods that operate entirely in the discrete domain, cDLS introduces a short excursion into continuous space to exploit gradient information more effectively.

### 4.1 Continuous Intermediates via Gradient Flow

Given the current discrete state $\theta \in \Theta$, we define a **continuous intermediate** $\theta^c$ by first taking a gradient ascent step in continuous space:

$$\theta^c_{\text{raw}} = \theta + \frac{\alpha_0}{2} \nabla U(\theta), \tag{6}$$

where $\alpha_0 > 0$ controls the magnitude of the exploration. Since $\theta^c_{\text{raw}}$ may be far from the feasible domain, we project it back into a convex region $K \subset \mathbb{R}^d$ that relaxes the original discrete domain $\Theta$:

$$\theta^c = \text{Proj}_K(\theta^c_{\text{raw}}) = \arg\min_{y \in K} \|y - \theta^c_{\text{raw}}\|. \tag{7}$$

This projected point $\theta^c$ serves as a **continuous intermediate** from which we construct a discrete proposal. $K$ is usually easy to decide. For example, if $\theta \in \{0,1\}^d$, then $K$ can be $[0,1]^d$. Such a projection is natural from the intrinsic structure of discrete spaces, ensuring that the continuous intermediates remain close to the feasible region. If $\theta_c$ moves substantially away from the initial discrete distribution space, this kind of update will be meaningless. The main purpose of this step is to inject gradient flow into the sampling process without permanently leaving the discrete space.

### 4.2 Discrete Langevin Proposal from Continuous Intermediates

Once the continuous intermediate $\theta^c$ is obtained, we construct a discrete proposal $\theta' \in \Theta$ using a Langevin-inspired method. We adopt the structure of the Discrete Langevin Proposal (DLP) (Zhang et al., 2022), but reinterpret it as centered at $\theta^c$:

$$q(\theta'|\theta^c) = \frac{\exp\left(-\frac{1}{2\alpha}\left\|\theta' - \theta^c - \frac{\alpha}{2}\nabla U(\theta^c)\right\|^2\right)}{Z_\Theta(\theta^c)}, \tag{8}$$

where $Z_\Theta(\theta^c)$ normalizes the distribution over the discrete domain $\Theta$. Since $\Theta$ is coordinate-factorized according to assumption, this proposal admits a product decomposition:

$$q(\theta'|\theta^c) = \prod_{i=1}^{d} q_i(\theta'_i|\theta^c), \tag{9}$$

where each $q_i$ is a categorical distribution defined as:

$$q_i(\theta'_i|\theta^c) \propto \exp\left(\frac{1}{2}\nabla U(\theta^c)_i(\theta'_i - \theta^c_i) - \frac{(\theta'_i - \theta^c_i)^2}{2\alpha}\right), \quad \theta'_i \in \Theta_i. \tag{10}$$

This form allows for parallelized sampling of each dimension and can be efficiently implemented. Crucially, by using $\theta^c$ instead of $\theta$, we incorporate richer gradient information and enable proposals that move beyond the local discrete neighborhood.

### 4.3 Metropolis-Hastings Correction

To ensure the resulting Markov chain has the desired stationary distribution $\pi(\theta)$, we optionally apply a Metropolis-Hastings (MH) correction (Metropolis et al., 1953; Hastings, 1970), which is usually combined with proposals to make the Markov chain reversible. Given the proposal $\theta' \sim q(\cdot|\theta^c)$, we compute the acceptance probability as:

$$A(\theta \to \theta') = \min\left(1, \exp\left(U(\theta') - U(\theta)\right) \cdot \frac{q(\theta|\theta'_c)}{q(\theta'|\theta^c)}\right), \tag{11}$$

where $\theta^c$ and $\theta'_c$ are the continuous intermediates corresponding to $\theta$ and $\theta'$, respectively. This correction is necessary when exact reversibility is required, although in practice, cDLS is often found to perform well even without MH correction due to its efficient utilization of local gradient information. Moreover, the MH correction provides an additional benefit of enabling adaptive stepsize selection based on the acceptance rate (Sun et al., 2022a), whereas samplers without MH correction typically rely on heuristic or grid-based tuning to identify suitable parameters.

### 4.4 Summary of the cDLS Framework

cDLS extends the capabilities of discrete Langevin samplers by introducing a gradient-guided continuous intermediate $\theta^c$ between discrete updates. This intermediate enables broader and more informed transitions while preserving the structure of discrete sampling. We outline the sampling algorithms in Algorithm 1. The different steps compared with Zhang et al. (2022) are highlighted in blue. We call the sampler with MH correction as *Discrete Metropolis Adjusted Langevin Algorithm via Continuous intermediates* (cDMALA) and without MH correction as *Discrete Unadjusted Langevin Algorithm via Continuous intermediates* (cDULA).

## 5 Convergence Analysis

In this section, we present a theoretical analysis of the proposed cDLS framework, including both the asymptotic convergence of cDULA and the non-asymptotic efficiency of cDMALA. Specifically:

- In Section 5.1, we show that for log-quadratic target distributions, cDULA introduces no asymptotic bias as the stepsize $\alpha \to 0$.

- In Section 5.2, we generalize this result to distributions that are approximately log-quadratic and characterize the bias in terms of deviation from linearity.

- In Section 5.3, we analyze cDMALA and prove its uniform ergodicity, providing guarantees on the convergence rate and statistical inference.

### 5.1 Convergence on Log-Quadratic Distributions

We begin with a special but fundamental case where the target distribution $\pi(\theta)$ has a log-quadratic form:

---

**Algorithm 1:** Discrete Langevin Samplers via Continuous Intermediates

**Input:** Stepsize $\alpha$, $\alpha_0$; Initial Sample $\theta_0$.
**while** *true* **do**
    **for** $i = 1, 2, \ldots, d$ **do**
        Compute $\theta^c_{raw} \leftarrow \theta + \frac{\alpha_0}{2}\nabla U(\theta)$;
        Project $\theta^c_{raw}$ as in Equation 7;
        Construct $q_i(\cdot|\theta^c)$ as in Equation 8;
        Sample $\theta'_i \sim q_i(\cdot|\theta^c)$;
    **end**
    // Optionally, do the MH step
    Compute $q(\theta'|\theta^c) = \prod_i q_i(\theta'_i|\theta^c)$;
    Compute $\theta^{c'}_{raw} \leftarrow \theta' + \frac{\alpha_0}{2}\nabla U(\theta')$;
    Project $\theta^{c'}_{raw}$ as in Equation 7;
    Compute $q(\theta|\theta'_c) = \prod_i q_i(\theta_i|\theta'_c)$;
    **if** *with MH step* **then**
        Set $\theta \leftarrow \theta'$ with probability in Equation 11;
    **else**
        Set $\theta \leftarrow \theta'$;
    **end**
**end**
**Output:** Samples $\{\theta_k\}$;

---

$$\pi(\theta) \propto \exp(\theta^\top W \theta + b^\top \theta), \quad \theta \in \Theta, \tag{12}$$

where $W \in \mathbb{R}^{d \times d}$ is assumed symmetric (otherwise replace with $(W + W^\top)/2$), and $b \in \mathbb{R}^d$. Such distributions frequently arise in structured models, like Ising model and Gaussian graphical models.

**Theorem 1.** *Assume the target distribution $\pi$ is log-quadratic as in Equation 12. Let $\lambda_{\min}$ be the smallest eigenvalue of $W$. Assume $\alpha_0 = o(\alpha)$ and consider any $\alpha > 0$, $\alpha_0 \geq 0$. Then:*

*i   The Markov chain defined by the proposal $q(\cdot|\theta)$ in Equation 8 is reversible.*

*ii   The stationary distribution $\pi_{\alpha,\alpha_0}$ of the chain converges weakly to $\pi$ as $\alpha \to 0$. Moreover, the convergence in L1 distance is bounded as:*

$$\|\pi_{\alpha,\alpha_0} - \pi\|_1 \leq cZ \cdot \exp\left( -\frac{1}{2\alpha} - \frac{(\alpha + \alpha_0)\lambda_{\min}}{2\alpha} \right),$$

*where $Z$ is the normalizing constant of $\pi$, and $c = \exp(\alpha_0 C \cdot D/4)$ depends on $\alpha_0$ and $D = \max_{\theta,\theta' \in \Theta} \|\theta' - \theta\|_\infty$.*

Theorem 1 shows that the bias of cDULA vanishes exponentially as $\alpha \to 0$, implying asymptotic correctness. Moreover, for any fixed $\alpha > 0$ and $\lambda_{\min} \geq 0$, the bound is strictly tighter than in the baseline case $\alpha_0 = 0$ (i.e., DULA), by an additional factor of $\exp\left(-\frac{\alpha_0 \lambda_{\min}}{2\alpha}\right)$, up to a mild multiplicative constant. Intuitively, the continuous intermediates steer the state toward high-probability regions before the discrete update, reducing the local linearization error that drives discretization bias under quadratic energies. In practice, this allows cDULA to maintain low bias even with moderate step sizes, reducing the need for fine-tuning and more iterations.

Figure 2(a) illustrates the behavior of cDULA with varying stepsizes on a $2 \times 2$ Ising model. As expected, the discrepancy between the stationary distribution and the target $\pi$ diminishes as $\alpha$ decreases. Additionally, cDULA demonstrates greater robustness compared to DULA under larger $\alpha$, validating the theoretical claim that continuous exploration mitigates bias and enables larger, more efficient steps.

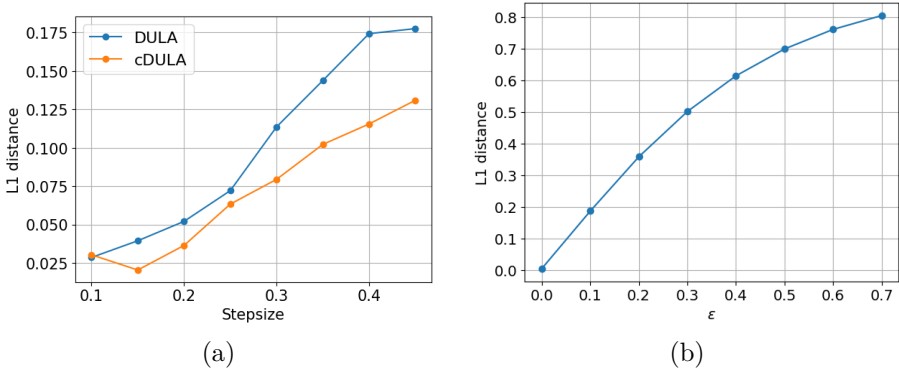

(a)                                                                 (b)

Figure 2: Verification of Theorem 1 and 2. (a) cDULA with varying stepsizes on $2 \times 2$ Ising model. (b) cDULA on 1-d distribution with varying closeness $\epsilon$ to being log-quadratic.

## 5.2   Convergence on General Distributions

We now extend the asymptotic analysis of cDULA to a broader class of target distributions. While Theorem 1 establishes exact convergence for log-quadratic targets, real-world energy functions often deviate from strict quadratic forms. To handle such cases, we quantify how closely the gradient of the energy function $U$ approximates a linear form.

Following Zhang et al. (2022), we assume there exists $W \in \mathbb{R}^{d \times d}$, $b \in \mathbb{R}^d$, and a small constant $\epsilon > 0$ such that

$$\|\nabla U(\theta) - (2W\theta + b)\|_1 \leq \epsilon, \quad \forall \theta \in \Theta. \tag{13}$$

This assumption states that the energy function is approximately log-quadratic in the gradient sense.

**Theorem 2.** *Let $\pi$ be the target distribution and $\pi'(\theta) \propto \exp(\theta^\top W\theta + b^\top\theta)$ be the log-quadratic approximation to $\pi$ as in Equation 13. Then the stationary distribution $\pi_{\alpha,\alpha_0}$ of cDULA satisfies:*

$$\|\pi_{\alpha,\alpha_0} - \pi\|_1 \leq 2c_1(\exp(c_2\epsilon) - 1) + Z'\exp\left(-\frac{1}{2\alpha} - \frac{(\alpha + \alpha_0)\lambda_{\min}}{2\alpha}\right),$$

*where $c_1$ depends on $\pi'$ and $(\alpha, \alpha_0)$, $c_2$ depends on $\Theta$ and $\max_{\theta,\theta'}\|\theta - \theta'\|_\infty$, and $Z'$ is the normalizing constant for $\pi'$.*

This result decomposes the total bias into two parts: The first term captures how far the true energy function $U$ is from a quadratic form, with error scaling exponentially in $\epsilon$. Thus, as $\epsilon \to 0$, the behavior of cDULA approaches that in Theorem 1; The second term is inherited from the previous theorem and quantifies the bias introduced by using non-zero stepsizes.

Together, these results offer practical guidance: to reduce bias, one should choose a continuous extension whose gradient is close to linear over the discrete support $\Theta$. To validate Theorem 2, we construct a 1D distribution of the form:

$$\pi(\theta) \propto \exp\left(a\theta^2 + b\theta + 2\epsilon\sin(\theta\pi/2)\right),$$

whose gradient is $\nabla U(\theta) = 2a\theta + b + \epsilon\pi\cos(\theta\pi/2)$. This gradient deviates from linearity in proportion to $\epsilon$. As shown in Figure 2(b), the total variation bias of cDULA increases with larger $\epsilon$, consistent with the theoretical bound. The trend closely matches that of DULA under the same perturbation, suggesting comparable robustness in mildly non-quadratic scenarios.

### 5.3 Non-asymptotic Convergence

Beyond asymptotic bias, we now analyze the non-asymptotic behavior of cDMALA. Our goal is to provide theoretical guarantees for the convergence rate and support statistical inference from finite samples. We establish these results by proving uniform ergodicity of the Markov chain $P$ under standard smoothness and local concavity assumptions, which are common in the literature on optimization and sampling (Dalalyan, 2017; Bottou et al., 2018).

**Assumption 1.** *The energy function $U(\theta) \in \mathcal{C}^2(\mathcal{R})$ has $L$-Lipschitz gradient. That is*

$$\|\nabla U(\theta) - \nabla U(\theta')\| \leq L\|\theta - \theta'\|. \tag{14}$$

**Assumption 2.** *For each $\theta \in \mathcal{R}^d$, there exists an open ball containing $\theta$ of some radius $r_\theta$, denoted by $B(\theta, r_\theta)$, such that the function $U(\cdot)$ is $m_\theta$-strongly concave in $B(\theta, r_\theta)$ for some $m_\theta > 0$.*

**Lemma 1.** *Let Assumptions 1 and 2 with $\alpha + \alpha_0 < \frac{2}{L}$ hold. Then for the Markov chain $P$ we have, for any $\theta, \theta' \in \Theta$,*

$$p(\theta|\theta') \geq \epsilon_{\alpha,\alpha_0}\frac{\exp(\tilde{\alpha}U(\theta'))}{\sum_{\theta'}\exp(\tilde{\alpha}U(\theta'))} \tag{15}$$

*where*

$$\epsilon_{\alpha,\alpha_0} = \exp\left\{-(\frac{2}{\alpha} + L - \frac{(\alpha + \alpha_0)m}{4\alpha})D^2 - \frac{\alpha + \alpha_0}{2\alpha}\|\nabla U(a)\|D\right\}.$$

**Theorem 3.** *Under Assumptions 1 and 2, and for $\alpha + \alpha_0 < \frac{2}{L}$, the Markov chain $P$ defined by cDMALA satisfies:*

*i $P$ is uniformly ergodic:*

$$\|P^n - \pi\|_{TV} \leq (1 - \epsilon_{\alpha,\alpha_0})^n.$$

*ii For any real-valued test function $f$, the sample average satisfies:*

$$\sqrt{n}\left(\frac{1}{n}\sum_{i=1}^{n}f(X_i) - \mathbb{E}_\pi[f]\right) \xrightarrow{d} \mathcal{N}(0, \sigma_*^2),$$

*for some $\sigma_* > 0$ as $n \to \infty$.*

Theorem 3 provides two levels of theoretical guarantees for cDMALA. First, it establishes *geometric ergodicity* of the Markov chain, i.e., the total variation distance between the marginal distribution after $n$ steps and the target distribution decays exponentially fast. Second, the Central Limit Theorem results in Theorem 3 provides an approach to perform inference on the target distribution $\pi$ even though the asymptotic variances are unknown, as we may perform batch-means to estimate these variances (Vats et al., 2019).

A key driver of this improvement lies in the modified parameter dependencies in the convergence bound. Compared to DMALA, the convergence coefficient $\epsilon_{\alpha,\alpha_0}$ of cDMALA features stronger contributions from both the gradient magnitude and the local curvature. Specifically, the gradient term is amplified by a factor of $(\alpha + \alpha_0)/2\alpha$, which enhances the directional bias of proposals toward high-probability regions when $\alpha_0$ is chosen such that $0 < \alpha_0 = o(\alpha)$. Meanwhile, the effective curvature term becomes $(\alpha + \alpha_0)m/4\alpha$, allowing the proposal to better exploit strong local concavity. Especially, under locally strongly concave potentials (large $m$) and when moderate stepsizes $\alpha$ are used, the effective lower bound $\epsilon_{\alpha,\alpha_0}$ of cDMALA is strictly larger than that of DMALA, which leads to faster convergence.

In summary, cDMALA is theoretically guaranteed to converge faster than DMALA whenever the target distribution exhibits sufficient local curvature and the auxiliary parameter $\alpha_0$ is scaled appropriately with $\alpha$. In these scenarios, the continuous intermediate acts as a curvature-aware regularizer, yielding more informed proposals and tighter non-asymptotic bounds..

# 6 Experiments

For both sampling tasks and learning tasks, we compare our method to Gibbs sampling, Gibbs-with-Gradient (GWG) (Grathwohl et al., 2021), Discrete Unadjusted Langevin Algorithm (DULA), Discrete Metropolis Adjusted Langevin Algorithm (DMALA) (Zhang et al., 2022), and Automatic Cyclical Scheduling (Pynadath et al., 2024), which are popular and recent gradient-based discrete samplers. All methods are implemented in `Pytorch`, and we use the official release of code from previous papers when possible. More experimental details can be found in Appendix H.

## 6.1 Sampling From Synthetic Distribution

To demonstrate the capability of cDLS to sample faster from the general distributions, we construct a 2-D energy landscape as follows:

$$U(\theta) = \log\left(\sum_{i=1}^{l} \exp\left(\frac{\|\theta - \mu_i\|^2}{2\sigma}\right)\right), \tag{16}$$

where $\Theta = \{0, 1, \cdots, N\}^2$, $N$ is the maximum value for each coordinate, and a set of modes $\{\mu_1, \mu_2, \cdots, \mu_l\}$. We demonstrate the results of various samplers in Figure 3, including Total Variance distance and visualization. More experimental details can be found in Appendix H.3.

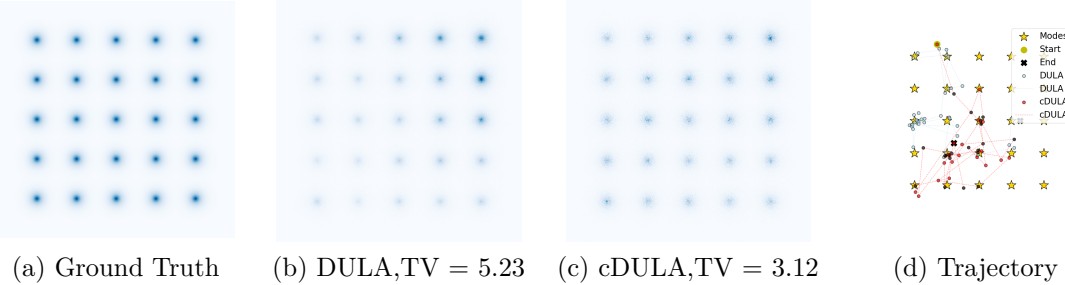

(a) Ground Truth     (b) DULA,TV = 5.23     (c) cDULA,TV = 3.12     (d) Trajectory

Figure 3: Synthetic Distribution Sampling. We run the DULA and cDULA sampler to compare how continuous exploration helps sample from the target distribution.

The results show that DULA focuses on several given modes, and by adding continuous exploration, the cDULA exhibits better performance. We also visualize the sampling trajectory of DULA versus cDULA in Figure 3 (d) and Appendix H.3, and it can be noticed that cDULA converges to the first mode very quickly by continuous exploration; in comparison, DULA often takes an excessively long sampling time.

## 6.2 Sampling From Ising Models

We consider a $5 \times 5$ lattice Ising model, where the random variable $\theta \in -1, 1^d$ with $d = 25$. The energy function is defined as $U(\theta) = a\theta^\top W\theta + b\theta$, where $W$ is the binary adjacency matrix of the lattice, $a = 0.1$ controls the connectivity strength, and $b = 0.2$ represents the external bias.

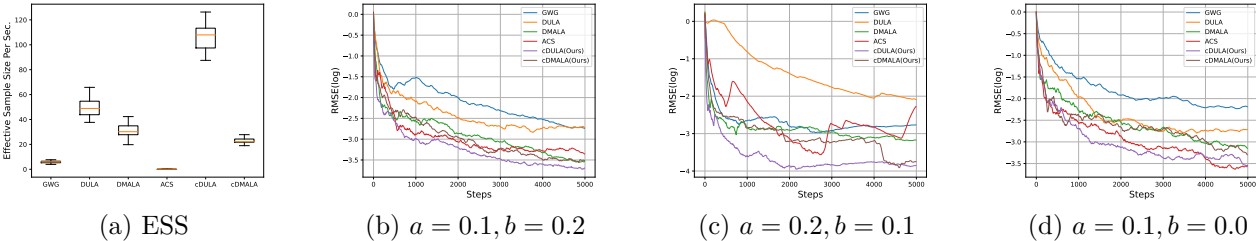

|   (a) ESS   |   (b) $a = 0.1, b = 0.2$   |   (c) $a = 0.2, b = 0.1$   |   (d) $a = 0.1, b = 0.0$   |

Figure 4: Ising model sampling results. (a) cDULA yields the largest effective sample size (ESS) per second and (b) achieves the lowest RMSE error among all the methods compared. (c)&(d)Also, our method maintains good performance despite the different parameters of the model. More discussion is detailed in the Appendix H.4.

**Results** As seen in Figure 4, cDULA significantly outperforms other methods. In Figure 4(a), cDULA yields the largest effective sample size (ESS) (Lenth, 2001) per second, indicating the correlation among its samples is low due to making significant updates in each step. We compare the root-mean-square error (RMSE) between the estimated mean and the true mean in Figure 4(b). This shows that exploration with continuous intermediates accelerates the convergence on this task, reaching the lowest or comparable RMSE error. In Figure 4(c)&(d), we see that cDULA significantly outperforms baseline methods in different settings. This suggests that cDULA effectively explores the continuous space to construct more informed proposals. These results demonstrate the ability to efficiently explore and accurately characterize modes. Notably, cDLS achieved satisfactory results even without MH correction. This may be attributed to the proposal distribution constructed from continuous intermediates, which fully leverages geometric information to perform well within a limited computational budget. However, for complex scenarios, such as Section 6.4.3, which tend to diverge, MH correction provides a strict guarantee of convergence.

## 6.3 Sampling From Restricted Boltzmann Machines

Building upon the previous experiments on the Ising model, we further evaluate our method on a more complex structured distribution, the Restricted Boltzmann Machine (RBM). RBMs are generative neural networks that learn an unnormalized probability distribution over inputs, defined as

$$U(\theta) = \sum_i \text{Softplus}(W\theta + a)_i + b^\top \theta, \tag{17}$$

where $\{W, a, b\}$ denote the model parameters and $\theta \in \{0, 1\}^d$ represents the visible binary variables. Following Grathwohl et al. (2021); Zhang et al. (2022), we initialize all samplers randomly, train the RBM parameters $\{W, a, b\}$ using contrastive divergence (Hinton, 2002), and evaluate sampling quality by computing the Maximum Mean Discrepancy (MMD) between the generated samples and those produced by Block-Gibbs sampling, which leverages the known structure of the RBM.

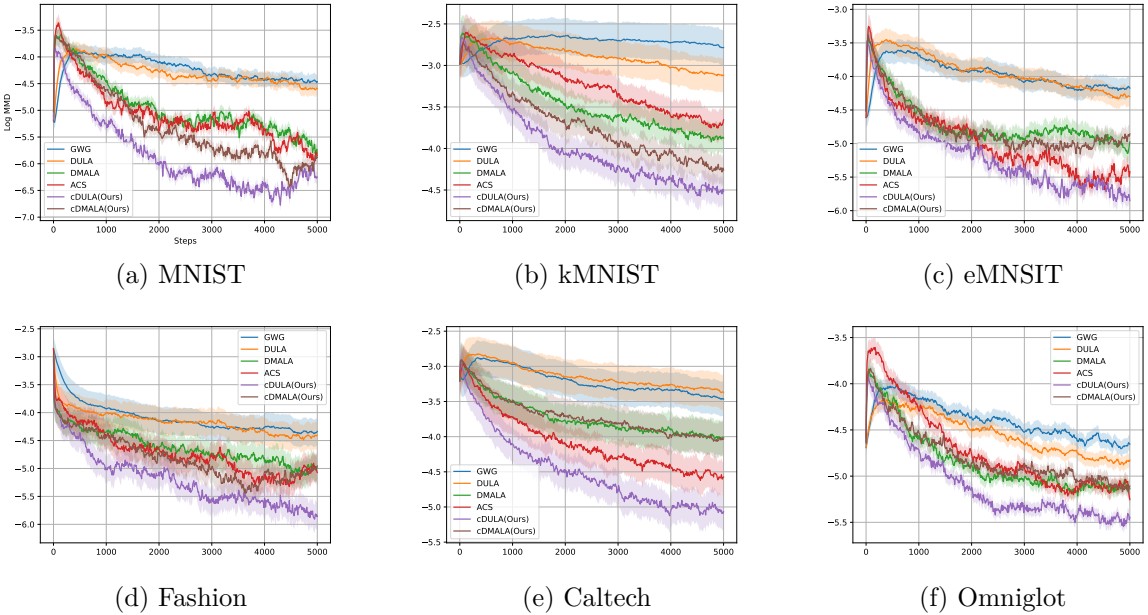

Figure 5: RBM models sampling results. Our method consistently achieves competitive or superior convergence across all datasets.

**Results** The results are shown in Figure 5. Our method, cDLS, especially cDULA, consistently achieves competitive or superior convergence across all datasets, with notable improvements on datasets like MNIST-like, Caltech and Omniglot. These results demonstrate the ability of the sampler to efficiently explore and accurately characterize modes. We further include the results of the generated images and runtime comparison in Appendix H.5.

## 6.4 Learning Energy-based Models

Energy-based models (EBMs) have demonstrated remarkable success across various machine learning domains (LeCun et al., 2006). An EBM defines a probability distribution as

$$f_\theta(x) = \frac{1}{Z_\theta} \exp(-E_\theta(x)) \tag{18}$$

where $E_\theta$ is a function parameterized by $\theta$ and $Z_\theta$ is the normalizing constant.[2] Training EBMs typically involves maximizing the log-likelihood,

$$\mathcal{L}(\theta) \overset{\triangle}{=} \mathbb{E}_{x \sim p_{data}}[\log f_\theta(x)]. \tag{19}$$

Since computing $Z_\theta$ directly is usually infeasible, the optimization instead relies on estimating the gradient of the log-likelihood (Song & Kingma, 2021):

$$\nabla \mathcal{L}(\theta) = \mathbb{E}_{x \sim p_\theta}[\nabla_\theta E_\theta(x)] - \mathbb{E}_{x \sim p_{data}}[\nabla_\theta E_\theta(x)]. \tag{20}$$

While the second expectation can be easily computed from data samples, the first requires drawing samples from the model distribution $p_\theta$, which is often the computational bottleneck. Consequently, the efficiency and quality of the sampler directly influence the training dynamics and the overall performance of EBMs.

---

[2]In the general formulation, the energy function appears as $\exp(-\beta E_\theta(x))$, where $\beta$ denotes the inverse temperature. In this work, following common practice in discrete sampling literature (Grathwohl et al., 2021; Zhang et al., 2022), we set $\beta = 1$ for simplicity without loss of generality.

### 6.4.1 Ising Models

Following Grathwohl et al. (2021); Zhang et al. (2022), we consider a $25 \times 25$ Ising model and generate data by running the Gibbs sampler. In this setting, the energy function $E_\theta$ corresponds to an Ising model parameterized by a learnable adjacency matrix $\hat{W}$. To evaluate the quality of different samplers, we compute the root mean squared error (RMSE) between the estimated $\hat{W}$ and the ground-truth $W$.

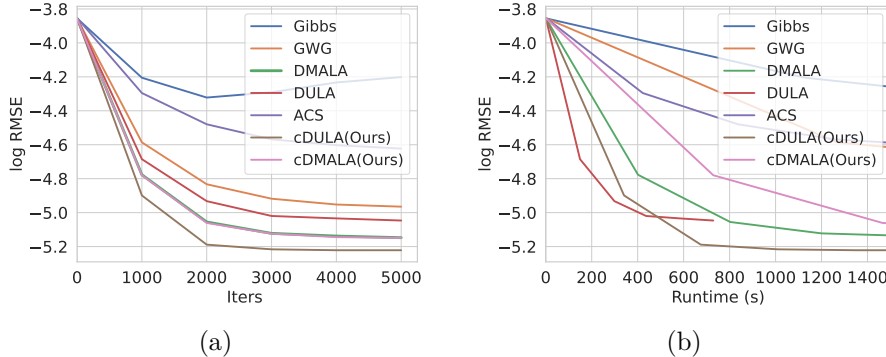

Figure 6: Ising model learning results. cDULA outperforms all baselines, finding the true W in a short time and getting a better convergence speed.

**Results**   Our results are summarized in Figure 6. cDULA outperforms all the compared methods, having the smallest RMSE among the baselines given the same number of iterations. Compared to the running time, cDULA also achieves the same log-RMSE in less time, except DULA as cDULA need time to explore the continuous space.

### 6.4.2 Learning RBM

We demonstrate the benefits of cDLS on RBMs. We evaluate the learned model using Annealed Importance Sampling (AIS) (Neal, 2001). We compare the sampling methods of interest to Block-Gibbs (BG), which utilizes the structure of RBMs well.

Table 1: Log likelihood scores for RBM learning on test data as estimated by AIS. Our method outperforms gradient-based baselines across most datasets. ACS results are mainly taken from Pynadath et al. (2024).

| Dataset | BG | GWG | DULA | DMALA | ACS | cDULA | cDMALA |
|---------|-----|-----|------|-------|-----|-------|--------|
| MNIST | *-208.76* | -372.59 | -271.17 | -234.43 | -249.55 | **-211.24** | -346.90 |
| eMNIST | *-337.90* | -445.68 | -312.12 | -349.85 | -304.96 | **-289.91** | -297.70 |
| kMNIST | *-347.72* | -519.72 | -381.27 | -452.43 | -407.39 | **-380.12** | -436.04 |
| Fashion | *-307.77* | -671.03 | -471.35 | -476.21 | -452.37 | **-406.30** | -460.93 |
| Omniglot | *-177.83* | -338.34 | **-180.66** | -267.83 | -220.71 | -190.47 | -281.47 |
| Caltech | *-524.34* | -593.47 | -538.50 | -731.18 | -396.04 | **-390.88** | -518.73 |

**Results**   From Table 1, we note that our methods outperform the baselines on most of the datasets. Since the structure of Omniglot is quite different from the other datasets (illustrated in Figure 12), it leads to insufficiency of exploration when using the same $\alpha_0$. This reveals that there should be a more adaptive approach to parameter tuning for different datasets, and we leave this for future work. We include more discussion of these experimental settings and the generated images in Appendix H.6.

Table 2: Log likelihood scores for EBM learning on test data as estimated by AIS. cDLS is able to achieve better results than the baselines. Previous results are mainly taken from Pynadath et al. (2024).

| Dataset | Gibbs | GWG | DULA | DMALA | ACS | cDULA | cDMALA |
|---|---|---|---|---|---|---|---|
| MNIST | -117.17 | -80.01 | -81.20 | -79.93 | -79.76 | -80.42 | **-79.55** |
| Dynamic MNIST | -121.19 | -80.51 | -80.06 | -80.13 | -79.70 | **-79.50** | -79.62 |
| Omniglot | -142.06 | -94.72 | -127.68 | -100.08 | -91.32 | -111.72 | **-91.01** |
| Caltech Silhouettes | -163.50 | -96.20 | -114.82 | -99.35 | -88.34 | -102.56 | **-87.82** |

Table 3: Experiment results with binary Bayesian neural networks on different datasets.

| Dataset | Training Log-likelihood ($\uparrow$) | | | | | | |
|---|---|---|---|---|---|---|---|
| | Gibbs | GWG | DULA | DMALA | ACS | cDULA | cDMALA |
| COMPAS | **-0.3102**$_{\pm 0.0029}$ | -0.3104$_{\pm 0.0022}$ | -0.3295$_{\pm 0.0049}$ | -0.3122$_{\pm 0.0011}$ | -0.3163$_{\pm 0.0002}$ | -0.3431$_{\pm 0.0001}$ | -0.3132$_{\pm 0.0033}$ |
| News | -0.2132$_{\pm 0.0013}$ | -0.2121$_{\pm 0.0023}$ | -0.2116$_{\pm 0.0013}$ | -0.2109$_{\pm 0.0001}$ | -0.2099$_{\pm 0.0071}$ | -0.2107$_{\pm 0.0001}$ | **-0.2098**$_{\pm 0.0033}$ |
| Adult | -0.3631$_{\pm 0.0007}$ | -0.3249$_{\pm 0.0010}$ | -0.3051$_{\pm 0.0001}$ | -0.2950$_{\pm 0.0042}$ | -0.3040$_{\pm 0.0014}$ | **-0.2900**$_{\pm 0.0002}$ | -0.3008$_{\pm 0.0030}$ |
| Blog | -0.3746$_{\pm 0.0021}$ | -0.3247$_{\pm 0.0004}$ | -0.2705$_{\pm 0.0023}$ | -0.2603$_{\pm 0.0031}$ | -0.2654$_{\pm 0.0043}$ | **-0.2601**$_{\pm 0.0006}$ | -0.2607$_{\pm 0.0009}$ |

| Dataset | Test RMSE ($\downarrow$) | | | | | | |
|---|---|---|---|---|---|---|---|
| | Gibbs | GWG | DULA | DMALA | ACS | cDULA | cDMALA |
| COMPAS | 0.4795$_{\pm 0.0034}$ | 0.4774$_{\pm 0.0037}$ | 0.4848$_{\pm 0.0013}$ | 0.4775$_{\pm 0.0029}$ | 0.4750$_{\pm 0.0039}$ | **0.4674**$_{\pm 0.0029}$ | 0.4789$_{\pm 0.0036}$ |
| News | 0.0964$_{\pm 0.0022}$ | 0.0975$_{\pm 0.0046}$ | 0.0994$_{\pm 0.0011}$ | 0.0946$_{\pm 0.0030}$ | 0.0992$_{\pm 0.0099}$ | 0.0932$_{\pm 0.0001}$ | **0.0905**$_{\pm 0.0009}$ |
| Adult | 0.4284$_{\pm 0.0022}$ | 0.4044$_{\pm 0.0004}$ | 0.3900$_{\pm 0.0016}$ | 0.3853$_{\pm 0.0003}$ | 0.3919$_{\pm 0.0072}$ | **0.3793**$_{\pm 0.0002}$ | 0.3869$_{\pm 0.0035}$ |
| Blog | 0.4040$_{\pm 0.0009}$ | 0.3564$_{\pm 0.0057}$ | 0.3186$_{\pm 0.0037}$ | 0.3130$_{\pm 0.0019}$ | 0.3168$_{\pm 0.0030}$ | **0.3077**$_{\pm 0.0040}$ | 0.3099$_{\pm 0.0004}$ |

### 6.4.3 Learning EBM

We further evaluate our methods on deep convolutional energy-based models (EBMs) to demonstrate their efficiency in large-scale settings. In these experiments, the energy function $E_\theta$ is parameterized by a ResNet (He et al., 2016), trained using Persistent Contrastive Divergence (PCD) (Tieleman, 2008; Tieleman & Hinton, 2009) with a replay buffer (Du & Mordatch, 2019), following the setups of Grathwohl et al. (2021) and Zhang et al. (2022). After training, we employ Annealed Importance Sampling (AIS) to estimate the model likelihood. Guided by the findings of Pynadath et al. (2024), we use 10 sampling steps per training iteration for all datasets, except for Caltech, where we adopt 30 steps to ensure sufficient mixing. Additional implementation details are provided in Appendix H.7.

**Results** Table 2 demonstrates that our method achieves superior performance over GWG and DLP. This advantage comes from our sampler's enhanced capacity to explore a broader range of modes per iteration, which accelerates the discovery of high probability regions. Such efficient exploration directly translates into higher-quality gradient estimates during training, driving more stable and effective model updates. It is reasonable to hypothesize that cDLS with multi-step or adaptive exploration steps will be more advantageous in such complex experiments (Loshchilov & Hutter, 2017; Zhang et al., 2020; Pynadath et al., 2024).

### 6.5 Binary Bayesian Neural Networks

Since MCMC-based methods are naturally suited for Bayesian inference, we further apply our approach to Bayesian neural networks (BNNs), where sampling efficiency plays a crucial role in model performance. Bayesian neural networks are known to provide reliable uncertainty estimation and strong predictive performance in deep learning (Hernández-Lobato & Adams, 2015; Zhang et al., 2020; Liu et al., 2021a). Meanwhile, binary neural networks (Courbariaux et al., 2016; Rastegari et al., 2016; Liu et al., 2021b), in which weights are restricted to $\{-1, 1\}$, offer significant computational and memory advantages. To combine the strengths of both paradigms, we study binary Bayesian neural networks trained via discrete sampling. We perform

regression experiments on four UCI datasets (Dua & Graff, 2017), where the energy function is defined as

$$U(\theta) = -\sum_{i=1}^{N} \|f_\theta(x_i) - y_i\|_2^2,$$

with $D = \{(x_i, y_i)\}_{i=1}^{N}$ denoting the training dataset, and $f_\theta$ representing a two-layer neural network with `Tanh` activation and 500 hidden units.

**Results**  From Table 3, we observe that our methods significantly outperform other gradient-based discrete samplers on all datasets except COMPAS, as there are fewer data samples. This phenomenon could also be found in Zhang et al. (2022). These results demonstrate that our methods converge fast for high-dimensional distributions, due to the ability to explore the parameter space using gradient, and suggest that our methods are compelling for training low-precision Bayesian neural networks where weights are discrete. Additionally, our method shows superior generalization for data, as RMSE on test datasets is significantly decreased while the training likelihood is almost equal.

## 6.6 Comparison of Variants

The primary advantage of cDULA over DULA comes from the possible regularization effect introduced by the continuous intermediates, which allows for maintaining the comparable error with a larger stepsize (under certain conditions, the discretization error will increase the stepsize), shown in Figure 2 (a). cDULA achieves a significant performance gain, remaining stable even with a doubled stepsize (i.e., $0.1 \to 0.2$), whereas DULA diverges quickly under the same stepsize.

When comparing cDULA and cDMALA, we observe that cDULA performs better in most of our experiments (Figure 5, Figure 6, Table 1). This improvement can be attributed to the absence of MH correction, which tends to reject certain proposals and thus results in more conservative updates. For training complex EBMs, however, this adventure action of cDULA can lead to divergence on challenging datasets (e.g., Caltech in Table 2), while cDMALA remains stable and converges more reliably — this stability is the main reason behind its strong performance. We present this trade-off between efficiency and robustness without extensive hyperparameter tuning to emphasize the inherent characteristics of cDULA and cDMALA.

The difference between cDMALA and DMALA entirely arises from the introduction of the continuous intermediates. Although MH correction in cDMALA occasionally leads to conservative updates, it still converges faster and yields larger effective sample sizes across several experiments (Figure 5, Table 5).

## 7  Conclusion

In our paper, we propose a hybrid discrete-continuous sampler for discrete spaces that extends gradient-based updates from purely discrete settings into an augmented continuous domain. By allowing samples to explore along gradient directions in this expanded space, our method improves the efficiency of sampling from high-dimensional discrete distributions. For different usage scenarios, we develop several corresponding variants, which include the unadjusted and Metropolis-Hastings adjusted versions. We prove the asymptotic convergence of our sampler under log-quadratic and general distributions and non-asymptotic bounds for the MH adjusted version. Numerous experiments on many different tasks show that our method outperforms the baseline methods. As a simple and easy-to-use method, we hope that our method will improve the sampling efficiency of discrete distributions and provide a new perspective on the utilization of gradient information.

There are still directions for further improvement of this domain. Currently, we adopt a fixed parameter $\alpha_0$ across all datasets for simplicity in a single experiment. However, energy landscapes often vary in smoothness and curvature, suggesting that an adaptive strategy for tuning $\alpha_0$ could better balance exploration and stability across different regimes. Although our methods already outperform existing baselines, designing more informed proposal mechanisms tailored to discrete domains remains a valuable direction, as such proposals could leverage structural priors to guide sampling more efficiently.

Moreover, our framework relies on the differentiability of the energy function, which may limit its applicability to certain discrete models. Relaxing this assumption or extending it to non-differentiable cases could further broaden the scope of our approach. Finally, even under differentiable settings, the current arrangement of discrete variables is heuristic; studying theoretical principles for optimal discrete variable organization may reveal deeper connections between geometry and efficiency, potentially leading to more principled and faster samplers.

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

## A    Algorithm with Binary Variables

cDLS could be simplified when the variable domain is binary $\Theta = \{0,1\}^d$. We obtain Algorithm 2 from 1 to help understand.

---

**Algorithm 2:** Discrete Langevin Samplers via Continuous Intermediates for Binary Variables

---

**Input:** The stepsize $\alpha, \alpha^0$

**while** *True* **do**

    Compute $\theta_{raw}^c = \theta + \frac{\alpha_0}{2}\nabla U(\theta)$;

    Project $\theta_{raw}^c$;

    Compute $P(\theta|\theta^c) = \dfrac{\exp\left(\frac{1}{2}\nabla U(\theta^c)\odot(1-\theta-\theta^c)-\frac{(1-\theta-\theta^c)^2}{2\alpha}\right)}{\exp(\frac{1}{2}\nabla U(\theta^c)\odot(1-\theta-\theta^c)-\frac{(1-\theta-\theta^c)^2}{2\alpha})+\exp(\frac{1}{2}\nabla U(\theta^c)\odot(\theta-\theta^c)-\frac{(\theta-\theta^c)^2}{2\alpha})}$;

    Sample $\mu \sim \text{Unif}(0,1)^d$;

    $I \leftarrow \mathbf{dim}(\mu \leq P(\theta|\theta^c))$;

    $\theta' \leftarrow \mathbf{flipdim}(I)$;

    // Optionally, do the MH step

    Compute $q(\theta'|\theta) = \prod_i q_i(\theta_i'|\theta) = \prod_{i\in I} P_i(\theta|\theta^c) \prod_{i\notin I}(1-P_i(\theta|\theta^c))$;

    Compute $\theta_{raw}^{c'} = \theta' + \frac{\alpha_0}{2}\nabla U(\theta')$;

    Project $\theta_{raw}^{c'}$;

    Construct $P(\theta'|\theta_c') = \dfrac{\exp\left(\frac{1}{2}\nabla U(\theta_c')\odot(1-\theta'-\theta_c')-\frac{(1-\theta'-\theta_c')^2}{2\alpha}\right)}{\exp(\frac{1}{2}\nabla U(\theta_c')\odot(1-\theta'-\theta_c')-\frac{(1-\theta'-\theta_c')^2}{2\alpha})+\exp(\frac{1}{2}\nabla U(\theta_c')\odot(\theta'-\theta_c')-\frac{(\theta'-\theta_c')^2}{2\alpha})}$;

    Compute $q(\theta|\theta') = \prod_i q_i(\theta_i|\theta) = \prod_{i\in I} P_i(\theta'|\theta_c') \prod_{i\notin I}(1-P_i(\theta'|\theta_c'))$;

    Set $\theta \leftarrow \theta'$ with probability in Equation 11;

**end**

**Output: samples**$\{\theta_k\}$;

---

## B    Algorithm with Categorical Variables

When using one-hot vectors to represent categorical variables, our algorithm becomes

$$\text{Categorical}\left(\text{Softmax}\left(\frac{1}{2}\nabla U(\theta^c)_i^\top (\theta_i' - \theta_i^c) - \frac{\|\theta_i' - \theta_i^c\|_2^2}{2\alpha}\right)\right).$$

where $\theta_i'$ is the one-hot vector, $\theta_c'$ is the explored vector. Notably, if one-hot vectors are used, the gradient-explored vectors should extend on the coordinate with one. That is say, Equation 7 for $\theta = e_i$ should follow that

$$\theta^c = \text{argmin}_{y\in K}||y - \theta c|| = Proj_K(\theta^c)$$

Actually, if the variables are ordinal with clear ordering information, we can also use integer representation $\theta \in \{0,1\cdots,K-1\}^d$ and follow Equation 6, Equation 7 and Equation 8 to sample from discrete distributions.

## C    Necessity of the Projection

One direct method is without Equation 7. That is, the sample has explored its domain by gradient, then updates according to the extended continuous sample. However, it may meet some obstacles and be beyond the sample domain, which will result in a very low acceptance probability in practice. For example, in the RBM experiment, its acceptance probability is 0.001% while cDMALA's acceptance probability is 62% on average. We use a simple illustration in Figure 1.

# D   Accelerating Other Gradient-Based Samplers via Continuous Intermediates

In the main text, we primarily focus on discrete Langevin samplers. In this section, we extend our framework to demonstrate how continuous intermediates can also accelerate other gradient-based discrete samplers.

Recall that in Grathwohl et al. (2021), Gibbs with Gradients (GWG) updates samples through several coordinate-wise steps. For simplicity, we consider the case of binary variables.

---

**Algorithm 3:** Gibbs With Gradients for Binary Variables

---

**Input:** energy function $U(\cdot)$, current sample $\theta$
**while** *True* **do**
    Compute $\tilde{d}(\theta) = -(2\theta - 1) \odot \nabla U(\theta)$;
    Compute $q(i|\theta) = \text{Categorical}\left(\text{Softmax}\left(\frac{\tilde{d}(\theta)}{2}\right)\right)$;
    Sample $i \sim q(i|\theta)$;
    $\theta' = \texttt{flipdim}(\theta, i)$;
    Compute $q(i|\theta') = \text{Categorical}\left(\text{Softmax}\left(\frac{\tilde{d}(\theta')}{2}\right)\right)$;
    Accept with probability:;
    $\min\left(\exp(U(\theta') - U(\theta))\frac{q(i|\theta')}{q(i|\theta)}, 1\right)$;
**end**
**Output: samples**$\{\theta_k\}$;

---

When computing $\tilde{d}(\theta)$, introducing a continuous intermediate enables the sampler to construct a more informed proposal. Specifically, we replace this step with

$$\theta^c = \text{Proj}_K(\theta + \frac{\alpha_0}{2}\nabla U(\theta)) = \arg\min_{y \in K} \|y - \theta - \frac{\alpha_0}{2}\nabla U(\theta)\|. \tag{21}$$

We refer to this variant as Gibbs with Gradients via Continuous intermediates (cGWG) and simply evaluate it on Restricted Boltzmann Machines (RBMs). Although cGWG updates only a single coordinate at each step, similar to the original GWG, it achieves consistently better performance, highlighting the effectiveness of incorporating continuous intermediates.

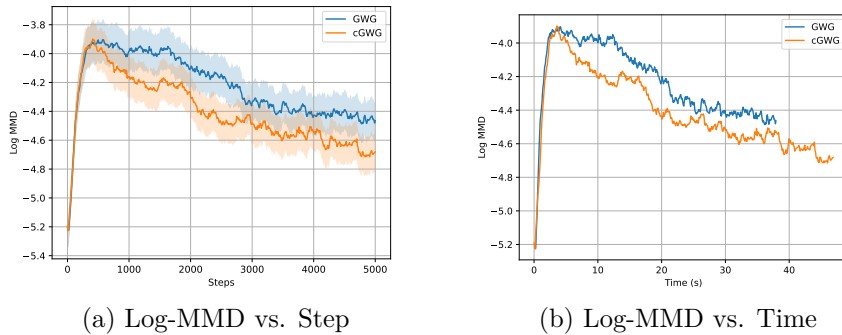

(a) Log-MMD vs. Step           (b) Log-MMD vs. Time

Figure 7: RBM Sampling for GWG and cGWG.

## E   Proof of Theorem 1

$$\theta_c = \theta + \frac{\alpha_0}{2}\nabla U(\theta)$$

$$\theta' \sim \text{Categorical}\left(\text{Softmax}\left(\frac{1}{2}\nabla U(\theta_c)^\top(\theta'-\theta_c) - \frac{||\theta'-\theta_c||_2^2}{2\alpha}\right)\right)$$

Expand the terms, we get

$$\frac{1}{2}\nabla U(\theta_c)^\top(\theta'-\theta - \frac{\alpha_0}{2}\nabla U(\theta)) - \frac{\|\theta'-\theta - \frac{\alpha_0}{2}\nabla U(\theta)\|_2^2}{2\alpha}$$

$$\overset{*}{=} \frac{1}{2}\nabla U(\theta)^\top(\theta'-\theta) - \frac{\|\theta'-\theta\|_2^2}{2\alpha} + (\frac{1}{2}\nabla U(\theta_c) - \frac{1}{2}\nabla U(\theta) + \frac{\alpha_0}{2\alpha}\nabla U(\theta))^\top(\theta'-\theta)$$

$$= \frac{\alpha+\alpha_0}{2\alpha}\nabla U(\theta)^\top(\theta'-\theta) - \frac{\|\theta'-\theta\|_2^2}{2\alpha} + \frac{\alpha_0}{4}\nabla U(\theta)^\top\nabla^2 U(\theta)(\theta'-\theta)$$

$$= \frac{\alpha+\alpha_0}{2\alpha}\nabla U(\theta)^\top(\theta'-\theta) - \frac{\|\theta'-\theta\|_2^2}{2\alpha} + \frac{\alpha_0}{4}\nabla U(\theta)^\top(\nabla U(\theta') - \nabla U(\theta))$$

$$\overset{*}{=} \frac{\alpha+\alpha_0}{2\alpha}\nabla U(\theta)^\top(\theta'-\theta) - \frac{\|\theta'-\theta\|_2^2}{2\alpha} + \frac{\alpha_0}{4}\nabla U(\theta)^\top(\nabla U(\theta'))$$

Here, $*$ means we drop some terms that won't affect the softmax result, i.e, not relevant to $\theta'$. Recall that the target distribution is $\pi(\theta) = \exp(\theta^\top W\theta + b^\top\theta)/Z$. We have that $\nabla U(\theta) = 2W^\top\theta + b, \nabla^2 U(\theta) = 2W$. Since $\nabla^2 U(\theta)$ is a constant, we can rewrite the proposal distribution as the following:

$$q_{\alpha,\alpha_0}(\theta'|\theta) = \frac{\exp\left(\frac{\alpha+\alpha_0}{2\alpha}\nabla U(\theta)^\top(\theta'-\theta) - \frac{\|\theta'-\theta\|_2^2}{2\alpha} + \frac{\alpha_0}{4}\nabla U(\theta)^\top(\nabla U(\theta'))\right)}{\sum_x \exp\left(\frac{\alpha+\alpha_0}{2\alpha}\nabla U(\theta)^\top(x-\theta) - \frac{\|x-\theta\|_2^2}{2\alpha} + \frac{\alpha_0}{4}\nabla U(\theta)^\top(\nabla U(x))\right)}$$

$$= \frac{\exp\left(\frac{\alpha+\alpha_0}{2\alpha}(U(\theta')-U(\theta)) - (\theta'-\theta)^\top(\frac{1}{2\alpha}I + \frac{\alpha+\alpha_0}{2\alpha}W)(\theta'-\theta) + \frac{\alpha_0}{4}\nabla U(\theta)^\top(\nabla U(\theta'))\right)}{\sum_x \exp\left(\frac{\alpha+\alpha_0}{2\alpha}(U(x)-U(\theta)) - (x-\theta)^\top(\frac{1}{2\alpha}I + \frac{\alpha+\alpha_0}{2\alpha}W)(x-\theta) + \frac{\alpha_0}{4}\nabla U(\theta)^\top(\nabla U(x))\right)}$$

Let $Z_{\alpha,\alpha_0}(\theta) = \sum_x \exp\left(\frac{\alpha+\alpha_0}{2\alpha}(U(x)-U(\theta)) - (x-\theta)^\top(\frac{1}{2\alpha}I + \frac{\alpha+\alpha_0}{2\alpha}W)(x-\theta) + \frac{\alpha_0}{4}\nabla U(\theta)^\top(\nabla U(x))\right)$, and $\pi_{\alpha,\alpha_0} = \frac{Z_{\alpha,\alpha_0}(\theta)\pi(\theta)}{\sum Z_{\alpha,\alpha_0}(x)\pi(x)}$, now we will show that $q_{\alpha,\alpha_0}$ is reversible w.r.t $\pi_{\alpha,\alpha_0}$. To simplify the notation, we use $\tilde{\alpha}$ to take place $(\alpha,\alpha_0)$.

We have that

$$\pi_{\tilde{\alpha}}q_{\tilde{\alpha}}(\theta'|\theta) = \frac{Z_{\tilde{\alpha}}(\theta)\pi(\theta)}{\sum Z_{\tilde{\alpha}}(x)\pi(x)} \cdot \frac{\exp\left(\frac{\alpha+\alpha_0}{2\alpha}(U(\theta')-U(\theta)) - (\theta'-\theta)^\top(\frac{1}{2\alpha}I + \frac{\alpha+\alpha_0}{2\alpha}W)(\theta'-\theta) + \frac{\alpha_0}{4}\nabla U(\theta)^\top(\nabla U(\theta'))\right)}{Z_{\tilde{\alpha}}}$$

$$= \frac{\exp\left(\frac{\alpha+\alpha_0}{2\alpha}U(\theta') + \frac{\alpha-\alpha_0}{2\alpha}U(\theta)) - (\theta'-\theta)^\top(\frac{1}{2\alpha}I + \frac{\alpha+\alpha_0}{2\alpha}W)(\theta'-\theta) + \frac{\alpha_0}{4}\nabla U(\theta)^\top(\nabla U(\theta'))\right)}{Z \cdot \sum_x Z_{\tilde{\alpha}}(x)\pi(x)}$$

Only and only if $\alpha_0 = o(\alpha)$, it is clear that this expression is symmetric in $\theta$ and $\theta'$. Therefore $q_{\tilde{\alpha}}$ is reversible and the stationary distribution is $\pi_{\tilde{\alpha}}$.

Now we will prove that $\pi_{\tilde{\alpha}}$ converges weakly to $\pi$ as $\alpha, \alpha_0 \to 0$ ($\alpha_0 = o(\alpha)$). Notice that for any $\theta$,

$$Z_{\tilde{\alpha}}(\theta) = \sum_x \exp\left(\frac{\alpha+\alpha_0}{2\alpha}(U(x)-U(\theta)) - (x-\theta)^\top(\frac{1}{2\alpha}I + \frac{\alpha+\alpha_0}{2\alpha}W)(x-\theta) + \frac{\alpha_0}{4}\nabla U(\theta)^\top(\nabla U(x))\right)$$

$$\overset{\tilde{\alpha}\downarrow 0}{=} \sum_x \exp\left(\frac{1}{2}(U(x)-U(\theta))\right)\delta_\theta(x)$$

$$= 1.$$

where $\delta_\theta(x)$ is a Dirac delta. Due to $Z_{\tilde\alpha}(\theta)$ is a continuous function when $\theta$ is fixed on $\mathcal{A} = \{\alpha > 0, \alpha_0 \geq 0\}$, so the limitation is well-defined. It follows that $\pi_{\tilde\alpha}$ converges pointwisely to $\pi(\theta)$. By Scheffe's Lemma, we attain $\pi_{\tilde\alpha}$ converges weakly to $\pi$.

**Convergence Rate w.r.t Stepsize** Let us consider the convergence rate in terms of $L_1$-norm.

$$\|\pi_{\tilde\alpha} - \pi\|_1 = \sum_\theta \left| \frac{Z_{\tilde\alpha}(\theta)\pi(\theta)}{\sum_x Z_{\tilde\alpha}(x)\pi(x)} - \pi(\theta) \right|.$$

We write out each absolute value term

$$\left| \frac{Z_{\tilde\alpha}(\theta)\pi(\theta)}{\sum_x Z_{\tilde\alpha}(x)\pi(x)} - \pi(\theta) \right| = \pi(\theta) \left| \frac{Z_{\tilde\alpha}(\theta)}{\sum_x Z_{\tilde\alpha}(x)\pi(x)} - 1 \right|$$

$$= \pi(\theta) \left| \frac{1 + \sum_{x\neq\theta} \exp(\frac{\alpha+\alpha_0}{2\alpha}(U(x)-U(\theta)) - (x-\theta)^\top(\frac{1}{2\alpha}I + \frac{\alpha+\alpha_0}{2\alpha}W)(x-\theta) + \frac{\alpha_0}{4}\nabla U(\theta)^\top(\nabla U(x) - \nabla U(\theta)))}{1 + \sum_y \frac{1}{Z}\exp(U(y)) \sum_{x\neq y} \exp(\frac{\alpha+\alpha_0}{2\alpha}(U(x)-U(\theta)) - (x-\theta)^\top(\frac{1}{2\alpha}I + \frac{\alpha+\alpha_0}{2\alpha}W)(x-\theta) + \frac{\alpha_0}{4}\nabla U(\theta)^\top(\nabla U(x) - \nabla U(\theta)))} - 1 \right|$$

where $\tilde\alpha = \frac{\alpha+\alpha_0}{2\alpha}$. Since $\lambda_{min}(W)\|x\|^2 \leq x^\top W x, \forall x$, it follows that

$$(x-\theta)^\top(\frac{1}{2\alpha}I + \frac{\alpha+\alpha_0}{2\alpha}W)(x-\theta) \geq \|x-\theta\|^2 (\frac{1+(\alpha+\alpha_0)\lambda_{min}}{2\alpha})$$

We also notice that $\min_{x\neq\theta}\|x-\theta\|^2 = 1$, and

$$\|\nabla U(\theta)^\top(\nabla U(x) - \nabla U(\theta))\|_2 = \|(2W^\top\theta + b)^\top(2W^\top(x-\theta)\|_2 \leq C\|x-\theta\|_\infty = CD$$

where $D = \max_{\theta,\theta'\in\Theta}\|\theta'-\theta\|_\infty$. Thus when $\frac{Z_{\tilde\alpha}(\theta)}{\sum_x Z_{\tilde\alpha}(x)\pi(x)} - 1 > 0$, we get

$$\left| \frac{Z_{\tilde\alpha}(\theta)\pi(\theta)}{\sum_x Z_{\tilde\alpha}(x)\pi(x)} - \pi(\theta) \right| = \pi(\theta) \left| \frac{Z_{\tilde\alpha}(\theta)}{\sum_x Z_{\tilde\alpha}(x)\pi(x)} - 1 \right|$$

$$= \pi(\theta) \left( \frac{1 + \sum_{x\neq\theta} \exp(\frac{\alpha+\alpha_0}{2\alpha}(U(x)-U(\theta)) - (x-\theta)^\top(\frac{1}{2\alpha}I + \frac{\alpha+\alpha_0}{2\alpha}W)(x-\theta) + \frac{\alpha_0}{4}\nabla U(\theta)^\top(\nabla U(x) - \nabla U(\theta)))}{1 + \sum_y \frac{1}{Z}\exp(U(y)) \sum_{x\neq y} \exp(\frac{\alpha+\alpha_0}{2\alpha}(U(x)-U(y)) - (x-y)^\top(\frac{1}{2\alpha}I + \frac{\alpha+\alpha_0}{2\alpha}W)(x-y) + \frac{\alpha_0}{4}\nabla U(y)^\top(\nabla U(x) - \nabla U(y)))} - 1 \right)$$

$$\leq \pi(\theta) \left( 1 + \sum_{x\neq\theta} \exp(\frac{\alpha+\alpha_0}{2\alpha}(U(x)-U(\theta)) - (\frac{1}{2\alpha}I + \frac{\alpha+\alpha_0}{2\alpha}W)\|x-\theta\|^2 + \frac{\alpha_0}{4}\nabla U(\theta)^\top(\nabla U(x) - \nabla U(\theta))) - 1 \right)$$

$$\leq \pi(\theta) \left( c \sum_x \exp(U(x)) \right) \cdot \exp\left( -\frac{1+(\alpha+\alpha_0)\lambda_{min}}{2\alpha} \right)$$

$$= c \cdot \pi(\theta) Z \cdot \exp(-\frac{1+(\alpha+\alpha_0)\lambda_{min}}{2\alpha})$$

where $c = \exp(\frac{\alpha_0}{4}CD)$ is a constant. Similarly, when $\frac{Z_{\tilde{\alpha}}(\theta)}{\sum_x Z_{\tilde{\alpha}}(x)\pi(x)} - 1 < 0$, we have,

$$
\left| \frac{Z_{\tilde{\alpha}}(\theta)\pi(\theta)}{\sum_x Z_{\tilde{\alpha}}(x)\pi(x)} - \pi(\theta) \right| = \pi(\theta) \left( 1 - \frac{1 + \sum_{x \neq \theta} \exp(\frac{\alpha+\alpha_0}{2\alpha}(U(x) - U(\theta)) - (x-\theta)^\top (\frac{1}{2\alpha}I + \frac{\alpha+\alpha_0}{2\alpha}W)(x-\theta) + \frac{\alpha_0}{4}\nabla U(\theta)^\top (\nabla U(x) - \nabla U(\theta)))}{1 + \sum_y \frac{1}{Z}\exp(U(y)) \sum_{x \neq y} \exp(\frac{\alpha+\alpha_0}{2\alpha}(U(x) - U(y)) - (x-y)^\top (\frac{1}{2\alpha}I + \frac{\alpha+\alpha_0}{2\alpha}W)(x-y) + \frac{\alpha_0}{4}\nabla U(y)^\top (\nabla U(x) - \nabla U(y)))} \right)
$$

$$
\leq \pi(\theta) \left( 1 - \frac{1}{1 + \sum_y \frac{1}{Z}\exp(U(y)) \sum_{x \neq y} c\exp(\frac{\alpha+\alpha_0}{2\alpha}(U(x) - U(\theta)) - \frac{1+(\alpha+\alpha_0)\lambda_{min}}{2\alpha})} \right)
$$

$$
= \pi(\theta) \left( \frac{\sum_y \frac{1}{Z}\exp(U(y)) \sum_{x \neq y} c\exp(\frac{\alpha+\alpha_0}{2\alpha}(U(x) - U(\theta)) - \frac{1+(\alpha+\alpha_0)\lambda_{min}}{2\alpha})}{1 + \sum_y \frac{1}{Z}\exp(U(y)) \sum_{x \neq y} c\exp(\frac{\alpha+\alpha_0}{2\alpha}(U(x) - U(\theta)) - \frac{1+(\alpha+\alpha_0)\lambda_{min}}{2\alpha})} \right)
$$

$$
\leq \pi(\theta) \left( \sum_y \frac{1}{Z}\exp(U(y)) \sum_{x \neq y} c\exp(\frac{\alpha+\alpha_0}{2\alpha}(U(x) - U(\theta))) \right) \cdot \exp\left( -\frac{1+(\alpha+\alpha_0)\lambda_{min}}{2\alpha} \right)
$$

$$
\leq \pi(\theta) \left( c\sum_x \exp(U(x)) \right) \cdot \exp\left( -\frac{1+(\alpha+\alpha_0)\lambda_{min}}{2\alpha} \right)
$$

$$
= c \cdot \pi(\theta) Z \cdot \exp(-\frac{1+(\alpha+\alpha_0)\lambda_{min}}{2\alpha})
$$

Therefore, the difference between $\pi_{\tilde{\alpha}}$ and $\pi$ can be bounded as follows.

$$
\|\pi_{\tilde{\alpha}} - \pi\| \leq c\sum_\theta \pi(\theta) Z \cdot \exp\left( -\frac{1+(\alpha+\alpha_0)\lambda_{min}}{2\alpha} \right) = cZ \cdot \exp\left( -\frac{1+(\alpha+\alpha_0)\lambda_{min}}{2\alpha} \right).
$$

## F   Proof of Theorem 2

*Proof* We use a log-quadratic distribution that is close to $\pi$ as an intermediate term to bound the bias of cDULA. Recall that $\pi$ is the target distribution, $\pi'$ is the log-quadratic distribution that is close to $\pi$ and $\pi_{\tilde{\alpha}}$ is the stationary distribution of cDULA. We let $\pi'_{\tilde{\alpha}}$ be the stationary distributions of cDULA targetting $\pi'$, then by triangle inequality,

$$
\|\pi_{\tilde{\alpha}} - \pi\|_1 \leq \|\pi_{\tilde{\alpha}} - \pi'_{\tilde{\alpha}}\|_1 + \|\pi'_{\tilde{\alpha}} - \pi'\|_1 + \|\pi' - \pi\|_1.
$$

**Bound of $\|\pi' - \pi\|_1$.**   Let the energy function of $\pi'$ be $V(\theta) = \theta^\top W\theta + b^\top \theta$. Since $\Theta$ is a discrete space, there exists a bounded subset $\Omega \in \mathrm{R}^d$ such that $\Theta$ is a subset of $\Omega$. By Poincar$\acute{e}$ inequality, we get

$$
|U(\theta) - V(\theta)| \leq C_1 \cdot \|\nabla U(\theta) - (W\theta + b)\|_1 \leq C_1\epsilon, \qquad \forall \theta \in \Theta
$$

where the constant $C_1$ depends on $\Omega$.

Recall that $\pi(\theta) = \frac{\exp(U(\theta))}{Z}$. Let $\pi'(\theta) = \frac{\exp(V(\theta))}{Z'}$ where $Z'$ is the normalizing constant to make $\pi'$ a distribution. Then

$$
\|\pi - \pi'\|_1 = \sum_{\theta \in \Theta} \left| \frac{\exp(U(\theta))}{Z} - \frac{\exp(V(\theta))}{Z'} \right|. \tag{22}
$$

We notice that $\forall \theta$,

$$
\frac{\exp(U(\theta))}{\exp(V(\theta))} = \exp(U(\theta) - V(\theta)) \leq \exp(|U(\theta) - V(\theta)|) \leq \exp(C\epsilon),
$$

and similarly

$$
\frac{\exp(V(\theta))}{\exp(U(\theta))} \leq \exp(C\epsilon).
$$

Therefore we have $\forall \theta$,

$$
\begin{aligned}
\left| \frac{\exp(U(\theta))}{Z} - \frac{\exp(V(\theta))}{Z'} \right| &= \left| \frac{\exp(U(\theta)) \cdot Z' - \exp(V(\theta)) \cdot Z}{Z \cdot Z'} \right| \\
&\leq (\exp(2C\epsilon) - 1) \min \left\{ \frac{\exp(U(\theta))}{Z}, \frac{\exp(V(\theta))}{Z'} \right\} \\
&\leq (\exp(2C\epsilon) - 1) \left( \frac{\exp(U(\theta))}{Z} + \frac{\exp(V(\theta))}{Z'} \right).
\end{aligned}
\tag{23}
$$

Plugging the above in Equation equation 22, we obtain

$$
\|\pi - \pi'\|_1 \leq 2 \left( \exp(2C\epsilon) - 1 \right).
$$

**Bound of $\|\pi'_{\tilde{\alpha}} - \pi'\|_1$.** By Theorem 1, we know

$$
\|\pi'_{\tilde{\alpha}} - \pi'\|_1 \leq cZ' \cdot \exp \left( -\frac{1 + (\alpha + \alpha_0)\lambda_{min}}{2\alpha} \right).
$$

**Bound of $\|\pi_{\tilde{\alpha}} - \pi'_{\tilde{\alpha}}\|_1$.** Now we will bound $\|\pi_{\tilde{\alpha}} - \pi'_{\tilde{\alpha}}\|_1$ by perturbation bounds. Let $T_{\tilde{\alpha}}$, $T'_{\tilde{\alpha}}$ be the transition matrices of GEDLS on $\pi$ and $\pi'$ respectively. We consider the difference between these two matrices.

$$
\left\| T_{\tilde{\alpha}} - T'_{\tilde{\alpha}} \right\|_\infty = \max_\theta \sum_{\theta'} \left| \frac{\exp \left( \frac{\alpha+\alpha_0}{2\alpha} \nabla U(\theta)^\top (\theta' - \theta) - \frac{\|\theta'-\theta\|_2^2}{2\alpha} + \frac{\alpha_0}{4} \nabla U(\theta)^\top (\nabla U(\theta')) \right)}{Z_\alpha(\theta)} - \frac{\exp \left( \frac{\alpha+\alpha_0}{2\alpha} \nabla V(\theta)^\top (\theta' - \theta) - \frac{\|\theta'-\theta\|_2^2}{2\alpha} + \frac{\alpha_0}{4} \nabla V(\theta)^\top (\nabla V(\theta'))}{Z'_\alpha(\theta)} \right. 
$$

where

$$
Z_\alpha(\theta) = \sum_x \exp \left( \frac{\alpha + \alpha_0}{2\alpha} \nabla U(\theta)^\top (\theta' - \theta) - \frac{\|\theta' - \theta\|_2^2}{2\alpha} + \frac{\alpha_0}{4} \nabla U(\theta)^\top (\nabla U(\theta')) \right),
$$

$$
Z'_\alpha(\theta) = \sum_x \exp \left( \frac{\alpha + \alpha_0}{2\alpha} \nabla V(\theta)^\top (\theta' - \theta) - \frac{\|\theta' - \theta\|_2^2}{2\alpha} + \frac{\alpha_0}{4} \nabla V(\theta)^\top (\nabla V(\theta')) \right).
$$

We denote $D = \max_{\theta,\theta' \in \Theta} \|\theta' - \theta\|_\infty$. By the assumption $\|\nabla U(\theta) - \nabla V(\theta)\|_1 \leq \epsilon$, we get

$$
\begin{aligned}
\exp \left( \nabla U(\theta)^\top \nabla U(\theta') - \nabla V(\theta)^\top \nabla V(\theta') \right) &\leq \exp \left( \left| \nabla U(\theta)^\top \nabla U(\theta') - \nabla V(\theta)^\top \nabla V(\theta') \right| \right) \\
&= \exp \left( \left| \nabla U(\theta)^\top \nabla U(\theta') - \nabla U(\theta)^\top V(\theta') + \nabla U(\theta)^\top V(\theta') - \nabla V(\theta)^\top \nabla V(\theta') \right| \right) \\
&\leq \exp \left( \left| \nabla U(\theta)^\top \nabla U(\theta') - \nabla U(\theta)^\top V(\theta') \right| + \left| \nabla U(\theta)^\top V(\theta') - \nabla V(\theta)^\top \nabla V(\theta') \right| \right) \\
&\leq \exp(M\epsilon)
\end{aligned}
$$

where $M = 2 \max_\theta \{ |\nabla V(\theta)| \}$. It gets from

$$
|\nabla U(\theta)| = |\nabla U(\theta) - \nabla V(\theta) + \nabla V(\theta)| \leq C_1 \epsilon + |\nabla V(\theta)|.
$$

Similar to Equation equation 23, we have

$$\left| \frac{\exp\left(\frac{\alpha+\alpha_0}{2\alpha}\nabla U(\theta)^\top(\theta'-\theta) - \frac{\|\theta'-\theta\|_2^2}{2\alpha} + \frac{\alpha_0}{4}\nabla U(\theta)^\top(\nabla U(\theta'))\right)}{Z_\alpha(\theta)} - \frac{\exp\left(\frac{\alpha+\alpha_0}{2\alpha}\nabla V(\theta)^\top(\theta'-\theta) - \frac{\|\theta'-\theta\|_2^2}{2\alpha} + \frac{\alpha_0}{4}\nabla V(\theta)^\top(\nabla V(\theta'))\right)}{Z'_\alpha(\theta)} \right|$$

$$\leq (\exp(N\epsilon) - 1)$$
$$\cdot \min\left\{ \frac{\exp\left(\frac{\alpha+\alpha_0}{2\alpha}\nabla U(\theta)^\top(\theta'-\theta) - \frac{\|\theta'-\theta\|_2^2}{2\alpha} + \frac{\alpha_0}{4}\nabla U(\theta)^\top(\nabla U(\theta'))\right)}{Z_\alpha(\theta)}, \frac{\exp\left(\frac{\alpha+\alpha_0}{2\alpha}\nabla V(\theta)^\top(\theta'-\theta) - \frac{\|\theta'-\theta\|_2^2}{2\alpha} + \frac{\alpha_0}{4}\nabla V(\theta)^\top(\nabla V(\theta'))\right)}{Z'_\alpha(\theta)} \right\}$$

$$\leq (\exp(N\epsilon) - 1)$$
$$\cdot \left( \frac{\exp\left(\frac{\alpha+\alpha_0}{2\alpha}\nabla U(\theta)^\top(\theta'-\theta) - \frac{\|\theta'-\theta\|_2^2}{2\alpha} + \frac{\alpha_0}{4}\nabla U(\theta)^\top(\nabla U(\theta'))\right)}{Z_\alpha(\theta)} + \frac{\exp\left(\frac{\alpha+\alpha_0}{2\alpha}\nabla V(\theta)^\top(\theta'-\theta) - \frac{\|\theta'-\theta\|_2^2}{2\alpha} + \frac{\alpha_0}{4}\nabla V(\theta)^\top(\nabla V(\theta'))\right)}{Z'_\alpha(\theta)} \right).$$

where $N = \frac{\alpha+\alpha_0}{\alpha}D + \frac{M\alpha_0}{2}$. Now we substitute it to $\left\|T_\alpha - \tilde{T}_\alpha\right\|_\infty$,

$$\left\|T - \tilde{T}\right\|_\infty \leq 2\left(\exp(N\epsilon) - 1\right).$$

By the perturbation bound in Schweitzer (1968),

$$\|\pi_{\tilde{\alpha}} - \pi'_{\tilde{\alpha}}\|_1 \leq C_2 \cdot \|T_{\tilde{\alpha}} - T'_{\tilde{\alpha}}\|_\infty = 2C_2\left(\exp(N\epsilon) - 1\right)$$

where $C_2$ is a constant depending on $\pi'$ and $\tilde{\alpha}$. Please note that it is also possible to use other perturbation bounds (Cho & Meyer, 2001).

Combining these three bounds, we get

$$\|\pi_\alpha - \pi\|_1 \leq \|\pi_\alpha - \pi'_\alpha\|_1 + \|\pi'_\alpha - \pi'\|_1 + \|\pi' - \pi\|_1$$
$$\leq 2C_2\left(\exp(N\epsilon) - 1\right) + cZ' \cdot \exp\left(-\frac{1 + (\alpha+\alpha_0)\lambda_{min}}{2\alpha}\right) + 2\left(\exp(2C_1\epsilon) - 1\right).$$

We define $c_1 := 2\max(2, 2C_2)$ and $c_2 := \max(2C_1, N)$, then we reach the the final result

$$\|\pi_\alpha - \pi\|_1 \leq \|\pi_\alpha - \pi'_\alpha\|_1 + \|\pi'_\alpha - \pi'\|_1 + \|\pi' - \pi\|_1$$
$$\leq 2c_1 \cdot \left(\exp(c_2\epsilon) - 1\right) + Z' \cdot \exp\left(-\frac{1 + (\alpha+\alpha_0)\lambda_{min}}{2\alpha}\right).$$

## G  Proof of Lemma 1 and Theorem 3

*Proof.* Recalling cDLS, we consider the proposal kernel as

$$q_{\tilde{\alpha}}(\theta'|\theta) \propto \exp\left\{ \tilde{\alpha}\nabla U(\theta)^\top(\theta'-\theta) - \frac{\|\theta-\theta'\|^2}{2\alpha} + \frac{1}{2}(\nabla U(\theta_c) - \nabla U(\theta))^\top(\theta'-\theta) \right\}$$

and consider the transition kernel as

$$p(\theta'|\theta) = \min\left\{ \frac{\pi(\theta')q_{\tilde{\alpha}}(\theta|\theta')}{\pi(\theta)q_{\tilde{\alpha}}(\theta'|\theta)}, 1 \right\} q_{\tilde{\alpha}}(\theta'|\theta) + (1 - L(\theta))\delta_\theta(\theta')$$

where $\delta_\theta(\theta')$ is the Kronecker delta function and $L(\theta)$ is the total acceptable probability from the point $\theta$ with

$$L(\theta) = \sum_{\theta'\in\Theta} \min\left\{ \frac{\pi(\theta')q_{\tilde{\alpha}}(\theta|\theta')}{\pi(\theta)q_{\tilde{\alpha}}(\theta'|\theta)}, 1 \right\} q_{\tilde{\alpha}}(\theta'|\theta).$$

We also define

$$Z_{\tilde{\alpha}}(\theta) = \sum_{x \in \Theta} \exp \left\{ \tilde{\alpha} \nabla U(\theta)^{\top} (x - \theta) - \frac{\|x - \theta'\|^2}{2\alpha} + \frac{1}{2} (\nabla U(\theta_c) - \nabla U(\theta))^{\top} (x - \theta) \right\}$$

which is the normalizing constant for the proposal kernel.

Consider the term,

$$\tilde{\alpha} \nabla U(\theta)^{\top} (\theta' - \theta) - \frac{1}{2\alpha} \|\theta - \theta'\|^2 = \tilde{\alpha}(-U(\theta) + U(\theta')) - \frac{1}{2}(\theta - \theta')^{\top} \left( \tilde{\alpha} \int_0^1 \nabla^2 U[(1-s)\theta + s\theta'] \mathrm{d}s + \frac{1}{\alpha} I \right) (\theta - \theta').$$

From Assumption 1, we have

$$\tilde{\alpha} \int_0^1 \nabla^2 U[(1-s)\theta + s\theta'] \mathrm{d}s + \frac{1}{\alpha} I \geq (\frac{1}{\alpha} - \tilde{\alpha} L) I.$$

Since $\alpha + \alpha_0 < \frac{2}{L}$, the matrix $(\frac{1}{\alpha} - \tilde{\alpha} L) I$ is positive define. We denote that

$$p(\theta'|\theta) = \min \left\{ \frac{\pi(\theta') q_{\tilde{\alpha}}(\theta|\theta')}{\pi(\theta) q_{\tilde{\alpha}}(\theta'|\theta)}, 1 \right\} q_{\tilde{\alpha}}(\theta'|\theta) + (1 - L(\theta)) \delta_\theta(\theta')$$

$$\geq \min \left\{ \frac{\pi(\theta') q_{\tilde{\alpha}}(\theta|\theta')}{\pi(\theta) q_{\tilde{\alpha}}(\theta'|\theta)}, 1 \right\} q_{\tilde{\alpha}}(\theta'|\theta)$$

$$\geq \min \left\{ \frac{Z_{\tilde{\alpha}}(\theta) \exp(\frac{\alpha_0}{4} L \|\nabla U(\theta')\|)}{Z_{\tilde{\alpha}}(\theta') \exp(\frac{\alpha_0}{4} LD \|\nabla U(\theta)\|)}, 1 \right\} q_{\tilde{\alpha}}(\theta'|\theta)$$

where $Z_{\tilde{\alpha}}$ could also be rewritten as

$$Z_{\tilde{\alpha}}(\theta) = \sum_{x \in \Theta} \exp \left\{ \tilde{\alpha} \nabla U(\theta)^{\top} (x - \theta) - \frac{\|x - \theta\|^2}{2\alpha} + \frac{1}{2} (\nabla U(\theta_c) - \nabla U(\theta))^{\top} (x - \theta) \right\}$$

$$= \sum_{x \in \Theta} \exp \left\{ \tilde{\alpha}(-U(\theta) + U(x)) - \frac{1}{2}(\theta - x)^{\top} \left( \tilde{\alpha} \int_0^1 \nabla^2 U[(1-s)\theta + sx] \mathrm{d}s + \frac{1}{\alpha} I \right) (\theta - x) + \frac{1}{2} (\nabla U(\theta_c) - \nabla U(\theta))^{\top} (x - \theta) \right\}$$

This can be seen as

$$\pi(\theta) q_{\tilde{\alpha}}(\theta'|\theta) = \frac{1}{Z Z_{\tilde{\alpha}}(\theta)} \exp \left\{ \tilde{\alpha}(U(\theta) + U(\theta')) - \frac{1}{2}(\theta - \theta')^{\top} \left( \tilde{\alpha} \int_0^1 \nabla^2 U[(1-s)\theta + s\theta'] \mathrm{d}s + \frac{1}{\alpha} I \right) (\theta - \theta') + \frac{1}{2} (\nabla U(\theta_c) - \nabla U(\theta))^{\top} (\theta' - \theta) \right\}.$$

Since Assumption 2 holds true in this setting, we have an $m > 0$ such that for any $\theta \in conv(\Theta)$

$$-\nabla^2 U(\theta) \geq mI.$$

From this, one notes that

$$\exp \left( -\tilde{\alpha} U(\theta) - \frac{1}{2}(\frac{1}{\alpha} - \tilde{\alpha} m) D^2 + \frac{\alpha_0}{4} L \|\nabla U(\theta)\| \right) \sum_{x \in \Theta} \exp(\tilde{\alpha} U(x)) \leq Z_{\tilde{\alpha}}(\theta) \leq \exp(-\tilde{\alpha} U(\theta) + \frac{\alpha_0}{4} LD \|\nabla U(\theta)\|) \sum_{x \in \Theta} \exp(\tilde{\alpha} U(x))$$

where the right-hand side follows from the fact that $\alpha + \alpha_0 < \frac{2}{L}$. Therefore,

$$\frac{Z_{\tilde{\alpha}}(\theta)}{Z_{\tilde{\alpha}}(\theta')} \geq \frac{\exp\{\tilde{\alpha}(-U(\theta) + U(\theta'))\} \cdot \exp\{\frac{\alpha_0}{4} L \|\nabla U(\theta)\|\}}{\exp\{\frac{1}{2}(\frac{1}{\alpha} - \tilde{\alpha} m) D^2\} \cdot \exp\{\frac{\alpha_0}{4} LD \|\nabla U(\theta')\|\}}$$

Also note that

$$q_{\tilde{\alpha}}(\theta'|\theta) = \frac{\exp \left\{ \tilde{\alpha}(-U(\theta) + U(\theta')) - \frac{1}{2}(\theta - \theta')^{\top} \left( \tilde{\alpha} \int_0^1 \nabla^2 U[(1-s)\theta + s\theta'] \mathrm{d}s + \frac{1}{\alpha} I \right) (\theta - \theta') + \frac{1}{2} (\nabla U(\theta_c) - \nabla U(\theta))^{\top} (\theta' - \theta) \right\}}{\sum_{x \in \Theta} \exp \left\{ \tilde{\alpha}(-U(\theta) + U(x)) - \frac{1}{2}(\theta - x)^{\top} \left( \tilde{\alpha} \int_0^1 \nabla^2 U[(1-s)\theta + sx] \mathrm{d}s + \frac{1}{\alpha} I \right) (\theta - x) + \frac{1}{2} (\nabla U(\theta_c) - \nabla U(\theta))^{\top} (x - \theta) \right\}}$$

$$\geq \frac{\exp \left\{ \tilde{\alpha} \nabla U(\theta)^{\top} (\theta' - \theta) - \frac{\|\theta - \theta'\|^2}{2\alpha} + \frac{1}{2} (\nabla U(\theta_c) - \nabla U(\theta))^{\top} (\theta' - \theta) \right\}}{\exp(-\tilde{\alpha} U(\theta) + \frac{1}{2} LD^2) \sum_{x \in \Theta} \exp(\tilde{\alpha} U(x))}$$

and

$$-\tilde{\alpha}\nabla U(\theta)^{\top}(\theta'-\theta) + \frac{\|\theta-\theta'\|^2}{2\alpha} - \frac{1}{2}(\nabla U(\theta_c) - \nabla U(\theta))^{\top}(\theta'-\theta) = \tilde{\alpha}(-\nabla U(\theta) + \nabla U(a))^{\top}(\theta'-\theta) - \nabla U(a)^{\top}(\theta'-\theta)$$

$$+ \frac{\|\theta-\theta'\|^2}{2\alpha} - \frac{1}{2}(\nabla U(\theta_c) - \nabla U(\theta))^{\top}(\theta'-\theta)$$

$$\leq \tilde{\alpha}\|\nabla U(a) - \nabla U(\theta)\| D + \tilde{\alpha}\|\nabla U(a)\| D + \frac{1}{2\alpha}D^2$$

$$\leq (\tilde{\alpha}L + \frac{1}{2\alpha})D^2 + \tilde{\alpha}\|\nabla U(a)\| D + \frac{1}{2}LD^2$$

Combining the inequities, we get

$$p(\theta'|\theta) \geq \epsilon_{\tilde{\alpha}} \frac{\exp\{\tilde{\alpha}U(\theta')\}}{\sum_{\theta'}\exp\{\tilde{\alpha}U(\theta')\}}$$

where

$$\epsilon_{\tilde{\alpha}} = \exp\left\{-(\frac{1}{\alpha} + \tilde{\alpha}L + L - \frac{\tilde{\alpha}m}{2})D^2 - \tilde{\alpha}\|\nabla U(a)\| D\right\}$$

To simplify, it also could be bound with $\alpha + \alpha_0 < \frac{2}{L}$,

$$\epsilon_{\tilde{\alpha}} = \exp\left\{-(\frac{1}{\alpha} + \tilde{\alpha}L + L - \frac{\tilde{\alpha}m}{2})D^2 - \tilde{\alpha}\|\nabla U(a)\| D\right\}$$

$$> \exp\left\{-(\frac{1}{\alpha} + \frac{2}{2\alpha \cdot L}L + L - \frac{\tilde{\alpha}m}{2})D^2 - \tilde{\alpha}\|\nabla U(a)\| D\right\}$$

$$= \exp\left\{-(\frac{2}{\alpha} + L - \frac{\tilde{\alpha}m}{2})D^2 - \tilde{\alpha}\|\nabla U(a)\| D\right\}.$$

As for Theorem 3, the proof directly follows from Lemma 1 and Cor.5 in Jones (2004).

## H    Additional Experiments Results and Setting Details

### H.1    Hyperparameter Settings

The key hyperparameters used for the compared methods are summarized as follows:

Table 4: Hyperparameter settings for all experiments.

| Sampler | GWG | DULA | DMALA | ACS | cDULA | cDMALA |
|---|---|---|---|---|---|---|
| **Table 1. General setup** | | | | | | |
| **Hyperparameter** | - | $\alpha$ | $\alpha$ | $[\beta_{\min}, \beta_{\max}]$ | $\alpha, \alpha_0$ | $\alpha, \alpha_0$ |
| **Table 2. Ising sample** | | | | | | |
| **Hyperparameter** | - | 0.2 | 0.4 | [0.5, 0.95] | 0.4, 0.2 | 0.4, 0.2 |
| **Table 3. RBM sample and learning** | | | | | | |
| **Hyperparameter** | - | 0.1 | 0.2 | [0.5, 0.95] | 0.2, 0.04 | 0.2, 0.04 |
| **Table 4. Ising learning** | | | | | | |
| **Hyperparameter** | - | 0.1 | 0.2 | [0.5, 0.95] | 0.2, 0.01 | 0.2, 0.01 |
| **Table 5. EBM learning** | | | | | | |
| **Hyperparameter** | - | 0.1 | 0.15 | [0.5, 0.95] | 0.15, 0.01 | 0.15, 0.01 |

We briefly introduce the hyperparameters used in different methods. GWG requires no additional hyperparameters, as it follows the Gibbs principle by updating one coordinate at a time based on gradient information. The parameter $\alpha$ in DLP and cDLS denotes the stepsize, analogous to that in continuous Langevin algorithms: a larger $\alpha$ leads to higher discretization error and potential divergence, whereas a smaller $\alpha$ results in more stable updates but requires more iterations for convergence. The parameter $\beta$ in ACS serves as a balancing parameter, where a cyclical schedule is employed to generate a sequence of $\{\beta_k\}$. Correspondingly, a heuristic procedure is used to determine the associated stepsizes $\{\alpha_k\}$. Further implementation details can be found in Pynadath et al. (2024). The $\alpha_0$ in cDLS controls the magnitude of exploration and is typically set not to exceed $\alpha$.

## H.2 Ising Model For Theory Verification

To verify Theorem 1, we use a 2 by 2 Ising model $U(\theta) = a\theta^\top W\theta + b\theta$ where $W$ is the binary adjacency matrix and $a = 0.1, b = 0.2$. To verify Theorem 2, we set $a = 1$ and $b = 0.1$.

## H.3 Sampling from Synthetic Distribution

We use the experiment setting of Pynadath et al. (2024). We need to define the space between the modes, the total number of modes, and the variance of each mode $\sigma$. For convenience, we take the same mode number in Pynadath et al. (2024), the space between modes as 15, and the variance for each mode $\sigma^2$ as .3. Based on this, we can calculate the maximum value for each coordinate as follows:

$$MaxVal = (\sqrt{NumModes} + 1) * SpaceBetweenModes.$$

The mode's center could be calculated as follows:

$$\mu_{i,j} = \frac{MaxVal}{\sqrt{NumModes} + 2}(i + 1)$$

$$\mu_{i,j} = \frac{MaxVal}{\sqrt{NumModes} + 2}(j + 1)$$

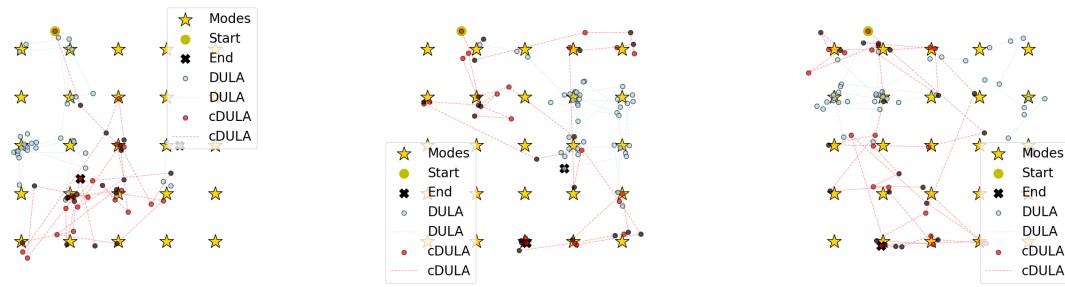

Figure 8: Sampling Trajectory with different random seeds.

## H.4 Sampling from Ising Models

We adopt the experiment setting of Zhang et al. (2022). The energy of the Ising model is defined as,

$$\log(p(x)) = ax^\top Jx + b^\top x$$

where $a$ controls the connectivity strength and $J$ is an adjacency matrix whose elements are either 0 or 1. If $J = 0$, then the model becomes a factorized Bernoulli distribution. In our experiments, $J$ is the adjacency matrix of a 2D lattice graph. cDULA and cDMALA use the same stepsize tuple (0.4,0.2). We run all methods for the same number of iterations. The results are shown in Figure 4. We can clearly observe that our methods outperform other baselines on all connectivity strengths. These three coupling strengths respectively correspond to real scenarios within the Ising model dominated by a strong external field, strong internal interactions, and the absence of an external field.

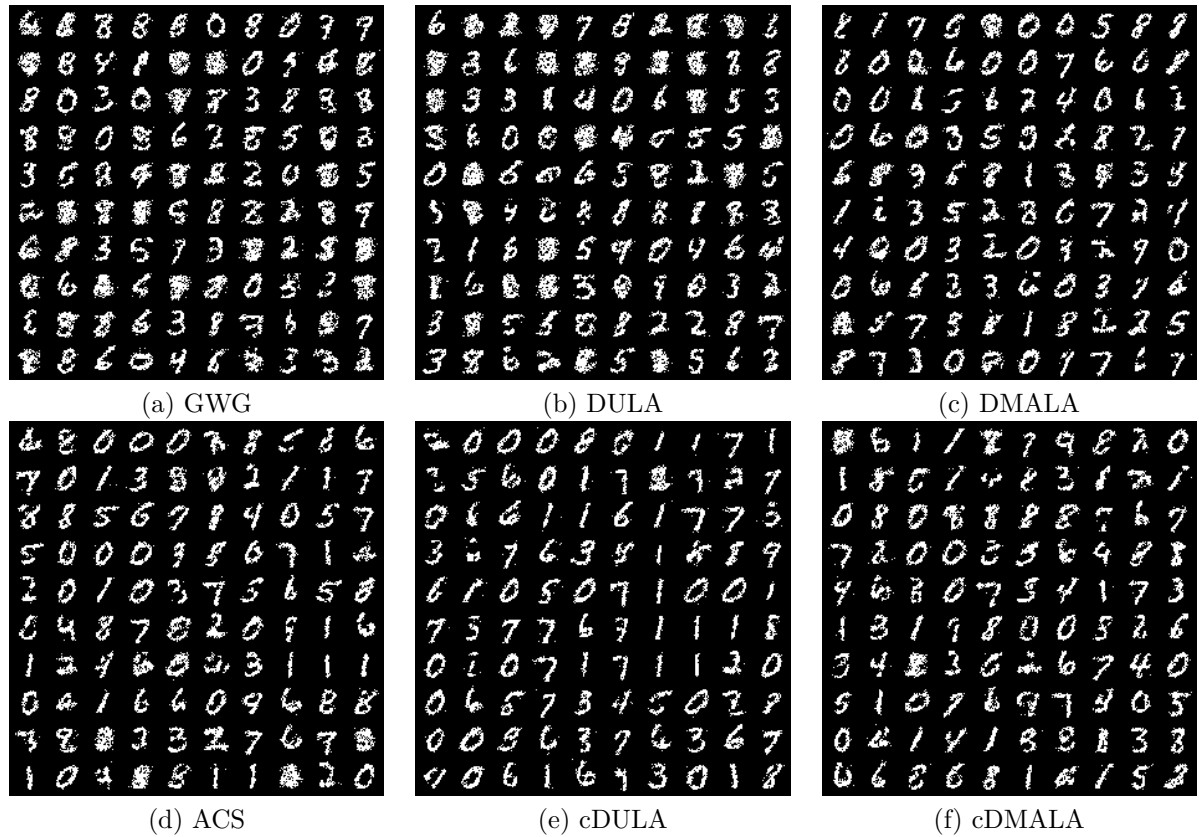

| (a) GWG | (b) DULA | (c) DMALA |
| (d) ACS | (e) cDULA | (f) cDMALA |

Figure 9: Generated samples from an RBM trained on MNIST. Our methods generate images that are closer to ground-truth samples.

## H.5  Sampling from RBMs

**RBM Overview**  We will give a brief overview of the Blocks-Gibbs sampler used to represent the ground truth of the RBM distribution. For a more in-depth explanation, see Grathwohl et al. (2021). Given the hidden units $h$ and the sample $x$, we define the RBM distribution as follows:

$$\log(p(x, h)) = h^\top W x + b^\top x + c^\top - \log(Z) \tag{24}$$

where $Z$ is the normalizing constant for the distribution. The sample $x$ is represented by the visible layer with units corresponding to the sample space dimension, and $h$ represents the model capacity. It can be shown that the marginal distributions are as follows:

$$p(x|h) = \text{Bernoulli}(Wx + c)$$
$$p(h|x) = \text{Bernoulli}(W^\top h + b)$$
$$\log(p(x)) = \sum_i \text{Softplus}(Wx + a)_i + b^\top x - \log(z).$$

The Block-Gibbs sampler updates $x$ and $h$ alternatively, allowing for many of the coordinates to get changes at the same time, due to utilizing the specific structure of the RBM model.

**Experiment Setup**   Similar to the experimental setup of Zhang et al. (2022); Pynadath et al. (2024), we use RBM models with 500 hidden units and 784 visible units. We adopt the same training protocol, except that we fine-tune the learning rate of Omniglot. From simpler MNIST-like data sets, we only train for a single pass through the dataset, while for more complicated datasets, we train for 3000 iterations to better represent the character and generate more realistic samplers. We include the generated images in Figure 12 to demonstrate that these models have learned the dataset reasonably well.

**Sampler Configuration**   For GWG, we use the same setting as Grathwohl et al. (2021). For DLP, we set the stepsize to 0.2 and 0.1 alternatively with MH correction and without MH correction. For cDLS, we use $\alpha = 0.2, \alpha_0 = 0.04$.

**Sampling Speed**   While the run time can vary depending on the specific implementation of a given sampling algorithm, we illustrate the efficiency of cDULA in Figure 10. cDULA is able to capture the efficiency of DMALA without Metropolis-Hastings correction, which significantly improves the convergence speed and generates more diverse samples.

**Effective sample size**   We find that the continuous exploration allows us to get closer to the high probability region before discrete sampling, allowing for larger steps of cDLS. In addition to accelerating the speed of convergence of the distribution, another significant benefit is the improvement in effective sample size (ESS). We run 100 chains in parallel with 5,000 iterations. Table 5 shows the ESS of the sampling of the RBM model, and it can be seen that our approach significantly improves the effective sample size.

Table 5: Effective sample size for RBM sampling. Our method outperforms gradient-based baselines across all datasets.

| Dataset | GWG | DULA | DMALA | ACS | cDULA | cDMALA |
|---------|-----|------|-------|-----|-------|--------|
| MNIST | $151.22_{\pm553.28}$ | $397.86_{\pm1051.08}$ | $676.07_{\pm1269.37}$ | $878.55_{\pm768.21}$ | $1948.68_{\pm1868.10}$ | $989.63_{\pm1518.10}$ |
| eMNIST | $143.49_{\pm513.91}$ | $312.11_{\pm939.11}$ | $262.66_{\pm755.76}$ | $478.32_{\pm372.79}$ | $1305.44_{\pm1802.20}$ | $587.28_{\pm1088.37}$ |
| kMNIST | $143.08_{\pm554.69}$ | $381.70_{\pm1023.72}$ | $518.35_{\pm1059.64}$ | $799.86_{\pm873.49}$ | $1601.81_{\pm1829.77}$ | $721.82_{\pm1218.88}$ |
| Fashion | $43.76_{\pm159.14}$ | $184.00_{\pm630.67}$ | $711.49_{\pm1232.28}$ | $977.23_{\pm1029.32}$ | $1357.86_{\pm1555.12}$ | $781.11_{\pm1387.46}$ |
| Caltech | $123.20_{\pm446.86}$ | $222.62_{\pm703.02}$ | $501.88_{\pm796.30}$ | $789.21_{\pm891.79}$ | $860.05_{\pm862.92}$ | $705.36_{\pm932.25}$ |
| Omniglot | $83.33_{\pm288.10}$ | $101.73_{\pm524.26}$ | $428.52_{\pm982.70}$ | $573.49_{\pm807.91}$ | $709.58_{\pm1395.64}$ | $504.40_{\pm1104.62}$ |

Table 6: Improvement on average acceptance probability.

| Dataset | DMALA | cDMALA | Dataset | DMALA | cDMALA |
|---------|-------|--------|---------|-------|--------|
| MNIST | 0.5307 | 0.6595 | Fashion | 0.5752 | 0.7025 |
| eMNIST | 0.5415 | 0.6706 | Caltech | 0.5339 | 0.7095 |
| kMNIST | 0.5307 | 0.6595 | Omniglot | 0.7430 | 0.8191 |

## H.6   Learning RBMs

**Experiment Design**   Follow Pynadath et al. (2024), We use the same RBM structure as the sampling task, with 500 hidden units and 784 visible units. However, we apply the samplers of interest to the PCD algorithm introduced by Tieleman (2008). The model parameters are tuned via the Adam optimizer Kingma & Ba (2015) with a learning rate of .001.

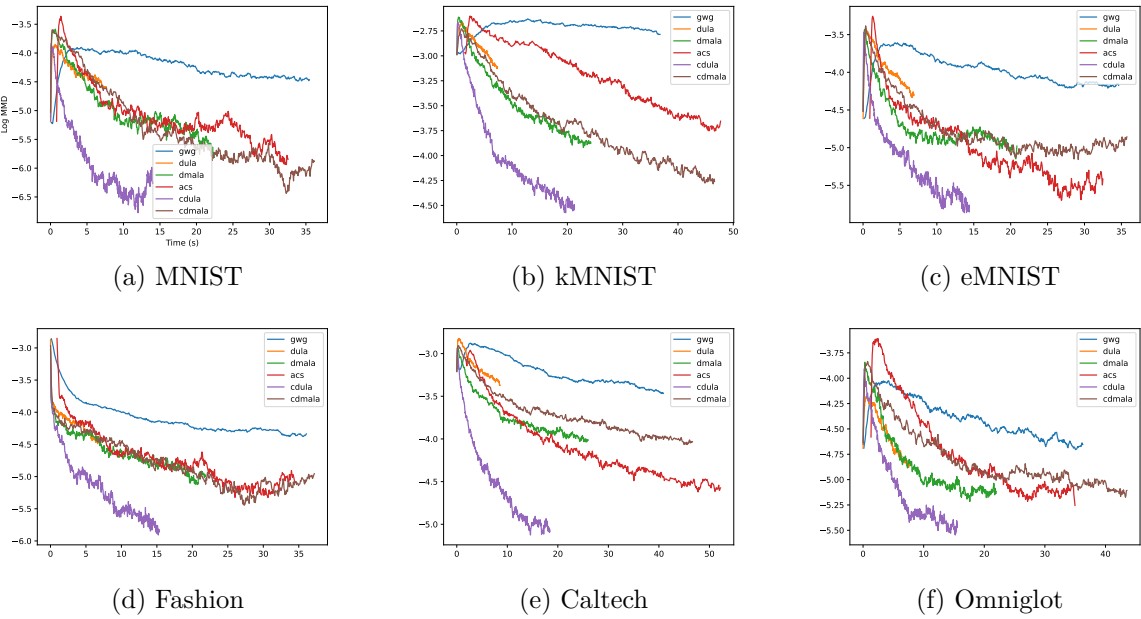

Figure 10: RBM Sampling with 5,000 sampling steps with results measured against time in seconds. cDULA is competitive both in terms of accuracy and efficiency.

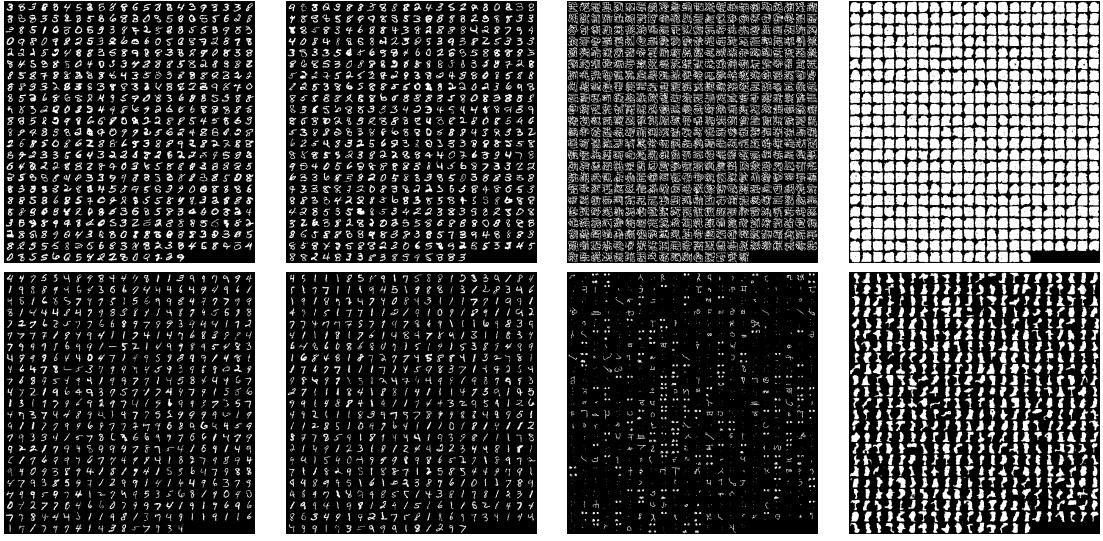

Figure 11: Left to right: Static MNIST, Dynamic MNIST, Omniglot, Caltech Silhouettes. Top: cDULA. Bottom: cDMALA.

In order to evaluate the learned RBMs, we run AIS with Block-Gibbs as the sampler to calculate the log likelihood values for the models Neal (2001). We run AIS for 100,000 steps, which is adequate given the efficiency of Block Gibbs for this specific model. The results are shown in Table 1.

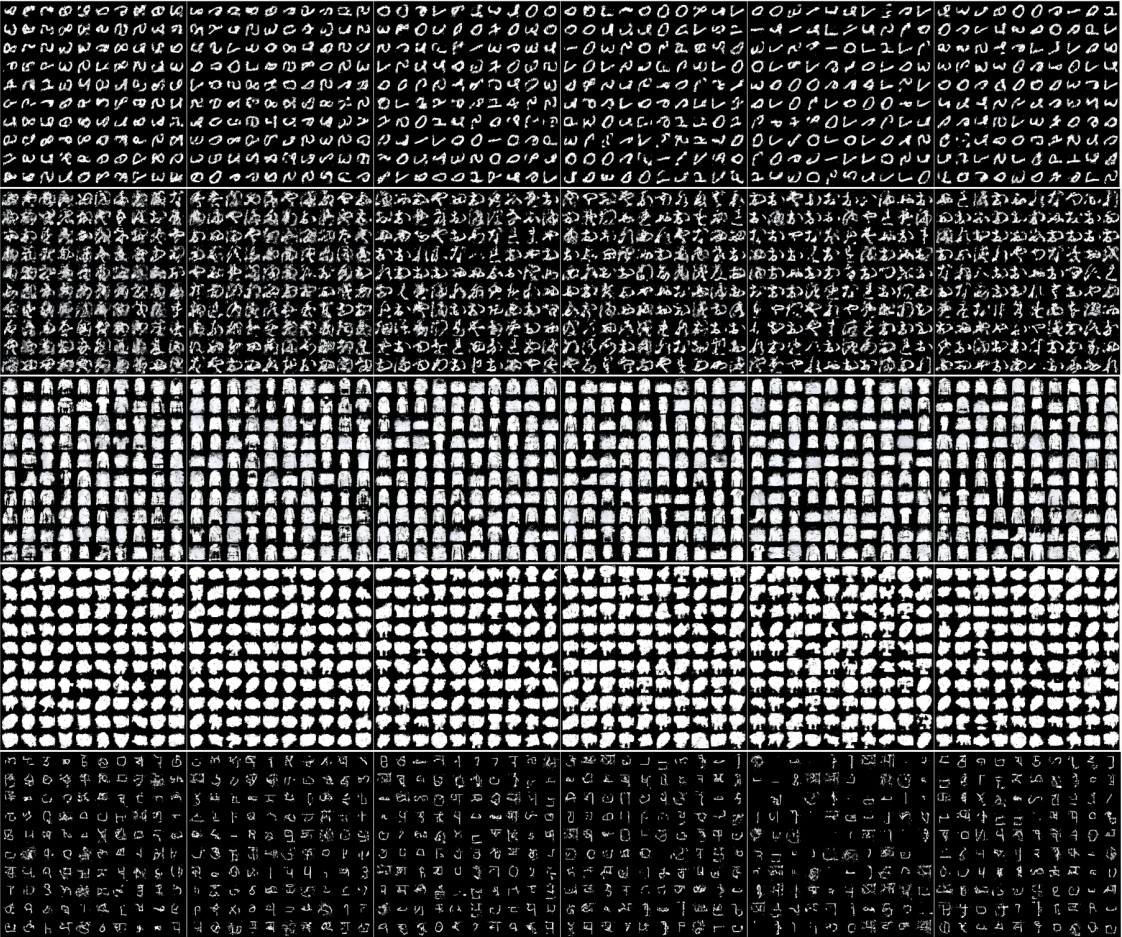

Figure 12: Images generated from RBMs trained by Constactive-Divergence with Block-Gibbs. Top to Bottom: eMNIST, kMNIST, Fashion, Caltech, Omniglot; Left to right: GWG, DULA, DMALA, ACS, cDULA and cDMALA.

### H.7 Learning EBMs

For all the experiments in this section, we use the stepsize $\alpha$ to be 0.1 for cDULA and 0.15 for cDMALA, $\alpha_0$ to be 0.01 for the two.

**Ising Model** We construct a training dataset of 2,000 instances by running 1,000,000 steps of Gibbs sampling for each instance. The model is trained by Persistent Constastive Divergence Tieleman (2008) with a buffer size of 256 samples. We also use the Adam optimizer with a learning rate of 0.001. The batchsize is 256. We train all models with an $l_1$ penalty with a penalty coefficient of 0.01 to encourage sparsity. The experiment setting is basically the same as Grathwohl et al. (2021); Zhang et al. (2022).

**Deep EBMs** We use the same EBM model architecture as Grathwohl et al. (2021); Zhang et al. (2022) and follow the same experimental design, where we implement ResNet with 8 residual blocks of 64 feature maps. We evaluate the models every 5,000 iterations by running AIS for 10, 000 steps. The reported results are from the model that performs the best on the validation set. The final reported numbers are generated by running 300, 000 iterations of AIS. All the models are trained with Adam with a learning rate of 0.0001 for 50, 000 iterations. We show the generated images with cDULA and cDMALA in Figure 11.

### H.8    Binary Bayesian Neural Networks

**Details of the Datasets**    (1) COMPAS: COMPAS J. Angwin & Kirchner (2016) is a dataset containing the criminal records of 6,172 individuals arrested in Florida. The task is to predict whether the individual will commit a crime again in 2 years. We use 13 attributes for prediction. (2) News: Online News Popularity Data Set [3] contains 39,797 instances of the statistics on the articles published by a website. The task is to predict the number of clicks in several hours after the articles are published. (3) Adult: Adult Income Dataset [4] is a dataset containing the information of US individuals from the 1994 census. The prediction task is to predict whether an individual makes more than 50K dollars per year. The dataset contains 44,920 data points. (4) Blog: Blog Feedback Buza (2014) is a dataset containing 54,270 data points from blog posts. The raw HTML documents of the blog posts were crawled and processed. The prediction task associated with the data is the prediction of the number of comments in the upcoming 24 hours. The feature of the dataset has 276 dimensions.

**Details on Training**    We run 10 chains in parallel and collect the samples at the end of training. All datasets are randomly partitioned into 80% for training and 20% for testing. The features and the predictive targets are normalized to (0,1). We set $\alpha = 0.1, \alpha_0 = 0.01$ for all datasets. We use a uniform prior over the weights. We train the Bayesian neural network for 2,000 steps. We use the full-batch training for the results in Section 6 so that Gibbs, GWG and DLP are also applicable. Beyond this, we could also get a stochastic variant for cDLS.

---

[3]https://archive.ics.uci.edu/ml/datasets/online+news+popularity
[4]https://archive.ics.uci.edu/ml/datasets/adult

