# OpenReview forum: "Accelerating Discrete Langevin Samplers via Continuous Intermediates"
_TMLR — Rejected by TMLR_

### Review · Reviewer_pQCq · 2025-10-16

**Summary Of Contributions:**

The paper introduces a new hybrid sampling approach for discrete domains that combines discrete updates with continuous intermediates to guide exploration. This framework enables gradient-informed exploration, aiming to accelerate convergence and improve sampling efficiency. The authors provide a thorough theoretical analysis and validate their approach with extensive experiments. The manuscript is well-written and easy to follow.

At the same time, I find the paper lacking in several aspects:

1. The novelty is somewhat understated and the work reads as an incremental extension of [1]. I think the manuscript should better highlight the novelty in the Introduction.
2. Claims about improved exploration are not clearly supported by experimental evidence. See "Requested Changes" below for details.
3. The paper does not discuss differences in performance between the proposed variants (cDULA vs. cDMALA) in depth. For example, considering Table 1 and Table 2, why in some cases cDULA outperforms cDMALA, and in others cDMALA outperforms cDULA?
4. Some performance patterns such as "DULA vs cDULA" and "DMALA vs cDMALA" should be discussed in detail. For example, when I look at Figure 5, I see that cDULA outperforms DULA more strongly, compared to cDMALA and DMALA performance difference. Why?

[1] Zhang et al. A Langevin-like sampler for discrete distributions. ICML, 2022.

**Audience:**

Yes

**Audience Explanation:**

I think the theoretical analysis provided in the paper will be useful for researchers working on discrete sampling, and they would likely find the approach and findings interesting.

**Broader Impact Concerns:**

No concerns.

**Claims And Evidence:**

Yes

**Claims Explanation:**

The theoretical claims are well-supported. The experimental results show that the proposed method outperforms baselines. However, the claim regarding improved exploration is not clearly evidenced. Including direct evidence in the main text would strengthen this claim.

**Requested Changes:**

1. The authors should clarify how their approach is novel rather than merely incremental in the Introduction. (Critical)
2. The authors should provide direct evidence that "the continuous intermediates improve exploration" because this is the underlying mechanism they claim for the performance improvement, but it is not experimentally shown in the main text. For example, they could move Figure 7 to the main text or add another experiment to demonstrate the exploration. (Critical)
3. Regarding Figure 7, what do the black circles represent? The authors should add a label to the figure legend. (Recommended)
4. The authors should discuss the performance differences between DULA, cDULA, DMALA, and cDMALA in detail. Please see points 3 and 4 in "Summary of Contributions" above for details. (Critical)
5. The authors should include a discussion in the Conclusion on potential limitations and directions for future work. (Recommended)

---

> ### Author Response · Authors · 2025-11-18
>
> We thank the reviewer for the thoughtful and constructive feedback. We address the main points below.
>
> **Q1: The authors should clarify how their approach is novel rather than merely incremental in the Introduction.**
>
> A1: We appreciate the comment from the reviewer. Our work indeed builds upon the discrete Langevin proposal (Zhang et al., 2022), but the underlying idea and implementation substantially differ. Specifically, we introduce a new continuous intermediate that accelerates discrete samplers by leveraging continuous augmented spaces. Although gradient-based samplers assume the existence of a natural continuous extension, they operate only
> in discrete spaces. our work is the first to operate in hybrid continous-discrete spaces.
> Importantly, our framework applies to any gradient-based discrete samplers [e.g.,GWG (Grathwohl et al., 2021)], not restricted to (Zhang et al., 2022). Although cGWG updates only a single coordinate at each step, similar to the original GWG, it achieves consistently better performance, highlighting the effectiveness of incorporating continuous intermediates. (Please see Appendix D) We will emphasize these aspects more clearly in the Introduction to highlight the distinct conceptual and methodological novelty of our approach.
>
> **Q2\&3: The authors should provide direct evidence that "the continuous intermediates improve exploration" because this is the underlying mechanism they claim for the performance improvement, $\cdots$, demonstrate the exploration.**
>
> A2\&3: We agree that the exploration advantage should be more directly shown. We have moved Figure 7 into the main text (Section 7.1). We also clarified that the black circles in Figure 7 represent continuous intermediates before discrete updates.
>
> **Q4: The authors should discuss the performance differences between DULA, cDULA, DMALA, and cDMALA in detail. Please see points 3 and 4 in "Summary of Contributions" above for details.**
>
> A4: We thank the reviewer for this insightful comment. Although comparisons among different variants were already discussed throughout the paper, appearing after each theorem and experiment, the scattered presentation may have obscured a unified analysis of their differences and connections. We have therefore added a subsection in Section 6 titled “Comparison of Variants” to analyze their performances under different scenarios.
>
> The primary advantage of cDULA over DULA comes from the continuous intermediates, which allows for maintaining the comparable error with a larger stepsize. In standard discrete Langevin samplers, the overall error is mainly governed by two factors: (1) discretization error, which decreases with a smaller stepsize; and (2) limited mixing per iteration, where a larger stepsize enables faster exploration but typically leads to divergence, especially in DULA. By incorporating continuous intermediates, cDULA is able to maintain discretization error comparable to (or even lower than) DULA while using a substantially larger stepsize, as illustrated in Figure 2 (a). A larger stepsize, in turn, allows the sampler to modify more coordinates per update, thereby resolving the issue that a smaller stepsize limits the change. This also generates more versatile samples (ESS). cDULA achieves a significant performance gain, remaining stable even with a doubled stepsize (0.1 → 0.2), whereas DULA diverges quickly under the same stepsize.
>
> When comparing cDULA and cDMALA, we observe that cDULA performs better in most of our experiments. This improvement can be attributed to the absence of MH correction, which tends to reject certain proposals and thus results in more 'conservative' updates. For training complex EBMs, however, this adventure action of cDULA can lead to divergence on challenging datasets (e.g., Caltech), while cDMALA remains stable and converges more reliably. This stability is the main reason behind its strong performance. We present this trade-off between efficiency and robustness without extensive hyperparameter tuning to emphasize the inherent characteristics of cDULA and cDMALA.
>
> The difference between cDMALA and DMALA entirely arises from the introduction of the continuous intermediates. Although MH correction in cDMALA occasionally leads to conservative updates, it still converges faster and yields larger effective sample sizes across several experiments.

---

> ### Author Response · Authors · 2025-11-18
>
> **Q5: The authors should include a discussion in the Conclusion on potential limitations and directions for future work.**
>
> A5: We sincerely thank the reviewer for the constructive feedback. We added a more concise paragraph in the Conclusion discussing future work. There are still directions for further improvement of this domain. Currently, we adopt a fixed parameter $\alpha_0$(recall that $\theta^c_{raw} = \theta + \frac{\alpha_0}{2}\nabla U(\theta)$) across all datasets for simplicity in a single experiment. However, energy landscapes often vary in smoothness and curvature, suggesting that an adaptive strategy for tuning could better balance exploration and stability across different regimes. Although our methods already outperform existing baselines, designing more informed proposal mechanisms tailored to discrete domains remains a valuable direction, as such proposals could leverage structural priors to guide sampling more efficiently.
>
> [1]Zhang et al. "A Langevin-like sampler for discrete distributions." ICML, 2022.
>
> [2]Grathwohl et al. "Oops I Took A Gradient: Scalable Sampling for Discrete Distributions." ICML, 2021.

---

> > ### Comment · Reviewer_pQCq · 2025-11-24
> >
> > I thank the authors for addressing my comments. I have no further questions.

---

### Review · Reviewer_DmDU · 2025-10-17

**Summary Of Contributions:**

This paper proposes Discrete Langevin Samplers via Continuous Intermediates (cDLS), a hybrid framework that accelerates sampling from high-dimensional discrete distributions. The core innovation is a short, continuous exploratory step between discrete updates, which allows the sampler to leverage gradient information for more informed, global transitions while preserving the discrete nature of the target space. The authors develop both unadjusted (cDULA) and Metropolis-adjusted (cDMALA) variants, providing strong theoretical guarantees of asymptotic correctness and non-asymptotic convergence. Extensive experiments on Ising models, RBMs, and deep energy-based models show that cDLS achieves faster convergence and higher sampling efficiency than existing methods.


### Strengths
- Presents an intuitive and novel bridge between discrete and continuous sampling, enabling more effective, gradient-informed global moves.
- The claims are supported by strong theoretical analysis, including both asymptotic correctness and non-asymptotic convergence guarantees.
- The framework is conceptually simple and can be readily incorporated into existing gradient-based discrete samplers.

### Weakness
- The method relies on a projection back into a convex region K that relaxes the discrete domain. The paper states that "K is usually easy to decide," but this assumption may not hold for more complex, non-standard discrete spaces, and the paper does not explore such cases.
- The additional cost of the continuous intermediate step is not fully characterized, which may be a practical limitation compared to the per-iteration cost of purely discrete methods.

**Audience:**

Yes

**Audience Explanation:**

Yes, this paper would strongly appeal to the TMLR audience. It tackles the crucial problem of efficient discrete sampling with a novel hybrid method that is both theoretically sound and empirically validated. The work's blend of rigorous convergence guarantees and demonstrated performance gains on relevant, modern machine learning models like EBMs and Bayesian Neural Networks  ensures it would be of significant interest to theorists, algorithm developers, and practitioners alike.

**Claims And Evidence:**

Yes

**Claims Explanation:**

Yes, the claims made in the submission are convincingly supported by both rigorous theory and extensive experiments. The paper's theoretical claims, including asymptotic correctness, bounded bias, and non-asymptotic convergence, are backed by formal proofs and directly verified with targeted numerical experiments that align with the theory's predictions . This theoretical foundation is complemented by comprehensive empirical evidence, where the proposed cDLS framework consistently outperforms strong baselines across a diverse range of applications.

**Requested Changes:**

- Analyze Computational Cost: The paper claims to "accelerate" sampling but does not adequately analyze the computational overhead of the continuous step. Please provide a more direct comparison of the wall-clock time versus iteration count to better quantify the practical efficiency trade-offs of the method.
- Clarify Scope of Projection: Please briefly discuss potential challenges or limitations in defining the projection step for more complex or structured discrete domains beyond those tested.
- Provide a Centralized Hyperparameter Table: To improve reproducibility, please consolidate the key hyperparameter settings for all compared methods into a single table.

---

> ### Author Response · Authors · 2025-11-18
>
> **Q1: Analyze Computational Cost**
>
> A1: We appreciate the reviewer’s insightful comment regarding the computational efficiency. The additional overhead of our method compared to DLP comes from computing the gradient of the continuous intermediates. For instance, in one iteration, cDULA performs one extra gradient computation compared to DULA. In practice, the actual runtime overhead depends on the underlying model architecture and implementation. To more directly quantify this trade-off, we have included logMMD-vs-Time figures (Fig.9) and average effective samples per second (Fig.4, Table 4) to evaluate both convergence and efficiency. Additionally, the following table summarizes representative wall-clock times (RBM on a single RTX3060), showing that cDLS achieves a favorable trade-off between computational cost and quality.
>
> | Sampler | GWG | DULA | DMALA | ACS | cDULA |cDMALA|
> |:----------:|:------:|:--------:|:--------:|:------:|:-----:|:----:|
> | Wall-clock Time(s) | 17.31 | 3.78 | 10.93 | 17.43 | 7.00 | 17.44 |
>
> **Q2: Clarify Scope of Projection**
>
> A2: We appreciate this important question about the limitations of the projection step. Our current focus is on accelerating discrete Langevin proposals where the objective function has an (implicit) differentiable form. In such cases, the projection K is quite easy to define — e.g., [0,1] for binary variables \{0,1\} (Section 4.1) and one-hot categorical variables (Appendix B). Indeed, it is non-trivial to find an appropriate projection for general discrete spaces due to the rugged and multimodal structure. This makes it difficult to directly exploit the underlying geometric information to design effective proposals.
>
> **Q3: Provide a Centralized Hyperparameter Table**
>
> A3: We appreciate the reviewer’s helpful suggestion. The key hyperparameters used for the compared methods are summarized as follows:
>
> ### Table 1. General setup
> | Sampler | GWG | DULA | DMALA | ACS | cDULA | cDMALA |
> |:----------:|:------:|:--------:|:--------:|:------:|:-----:|:----:|
> | Hyperparameter | - | $\alpha$ | $\alpha$ | [$\beta_{min}, \beta_{max}$] | $\alpha, \alpha_0$ | $\alpha, \alpha_0$ |
>
> ---
>
> ### Table 2. Ising sample
> | Sampler | GWG | DULA | DMALA | ACS | cDULA | cDMALA |
> |:----------:|:------:|:--------:|:--------:|:------:|:-----:|:----:|
> | Hyperparameter | - | 0.2 | 0.4 | [0.5, 0.95] | 0.4, 0.2 | 0.4, 0.2 |
>
> ---
>
> ### Table 3. RBM sample and learning
> | Sampler | GWG | DULA | DMALA | ACS | cDULA | cDMALA |
> |:----------:|:------:|:--------:|:--------:|:------:|:-----:|:----:|
> | Hyperparameter | - | 0.1 | 0.2 | [0.5, 0.95] | 0.2, 0.04 | 0.2, 0.04 |
>
> ---
>
> ### Table 4. Ising learning
> | Sampler | GWG | DULA | DMALA | ACS | cDULA | cDMALA |
> |:----------:|:------:|:--------:|:--------:|:------:|:-----:|:----:|
> | Hyperparameter | - | 0.1 | 0.2 | [0.5, 0.95] | 0.2, 0.01 | 0.2, 0.01 |
>
> ---
>
> ### Table 5. EBM learning
> | Sampler | GWG | DULA | DMALA | ACS | cDULA | cDMALA |
> |:----------:|:------:|:--------:|:--------:|:------:|:-----:|:----:|
> | Hyperparameter | - | 0.1 | 0.15 | [0.5, 0.95] | 0.15, 0.01 | 0.15, 0.01 |

---

### Review · Reviewer_YgnU · 2025-10-27

**Summary Of Contributions:**

The authors develop a new method to approach the challenging problem of sampling from discrete distributions. The method builds on recently proposed discrete Langevin samplers and uses gradient information through a continuous relaxation + projection into a bounded region within each discrete update. Overall, this may be a promising paper, but the current state is poorly written, at times incoherent, and difficult to follow. One absolutely has to read the original 2022 paper (Zhang et al., 2022) to understand what the authors are trying to improve upon and wherein the contribution lies. Moreover, many paragraphs and large portions of the theory is taken **verbatim** from said paper, which is concerning. Concretely:

- The Introduction (and the whole paper) can benefit from a more coherent flow that does not jump from topic to topic in every sentence. For instance, already the first paragraph is a mixture of LLM-tuned text and short sentences that do not connect well. Lots of jargon is used which is familiar to MCMC practitioners and developers but may not be standard vernacular for ML researchers.

- The mathematical formulation is inconsistent. Discrete target distributions are typically expressed using a negative energy ($\exp(-\beta E)$), not $\exp(E)$, which implicitly assumes a negative temperature set to 1. This sign inconsistency appears throughout the paper and undermines the connection to energy-based modeling.

- Similarly to the Introduction, the Preliminaries do not flow very well, contain typos, and feature wrong equation references (e.g., eq. 3 is referenced before eq. 1). Moreover, it would be helpful if the paragraphs in the preliminaries are connected to the current work instead of ending on a general LLM-tuned sentence (e.g., what are the disadvantages of Langevin sampling and why isn’t it adopted by the Bayesian community as compared to HMC - just to give an example). Also, it is not clear in which sense the DLP is a “counterpart of the Langevin algorithm in discrete domains”. It somehow involves a differentiable energy function with respect to the discrete random vector which already assumes some differentiable extension but uses the same symbol.

- Having read Section 4, I am left puzzled as to what exactly is the innovation of the current paper apart from setting out of bounds relaxed realizations of $\theta$ to the closest point on a hypergrid (i.e., Eq. 6). The projection into a bounded space seems trivial and I am surprised this isn’t something that DLP already does in practice (just as HMC never learns in bounded spaces in practice). Also, it is unclear how the relaxation and back-projection happen in DLP and, consequently, in the current method. It is also unclear how to sample from the distribution defined in Eq.7. The text tends to oscillate between heuristic interpretation and formalism without committing to either.

- I failed to see how applying the MH correction (Section 4.3) does not inherit all limitations (e.g., curse of dimensionality) of basic MH MCMC. Additionally, the correction is introduced without any motivation or justification. - Algorithm 1 deviates from the notation in the text (e.g., does not use the $\theta_{raw}$ subscript), uses a $k$ index for samples that is only implied, and adopts a peculiar language to describe the steps (Explore -> this is rather an update step; map -> this should be relaxation + projection, etc)

- The results do seem encouraging and the authors clearly put quite some effort into the experimental validation. However, the descriptions of the experiments are hardly possible to follow and the Appendix does little to help, as all of it reads more like lab notes than text. Already Figure 3 does not match the description of the text and is hard to interpret. Why are 25 modes necessary and why are the samples from the relaxed distribution shown? Without further information, it seems that both samplers suffer from mode collapse. There seem to be errors in the figures, for instance, DMALA is present in the legend of Figure 6 but nowhere to be seen. Generally, I believe it may be helpful to extend the paper beyond 12 pages and may delegate an entire experiment to the Appendix (e.g., 6.3, which seems to merely compare several samples qualitatively).

- The same goes for the theoretical results.  Admittedly, I did not go over all the math, but there are obvious typos in the Appendix (e.g., matrix - vector multiplications not matching dimensions) and barely any structure to hold on to. Much of the theory seems to be taken **verbatim** from Zhang et al. (2022). Theorem 2 appears to simply restate Theorem 1 under weakly perturbed conditions.

- The connection between the main results and appendices is extremely weak. The appendix is bloated with raw formulas and code-like text but fails to clarify key implementation details or parameter choices, making reproducibility impossible.

**Audience:**

Yes

**Audience Explanation:**

Sampling from discrete distributions is an important problem in machine learning and statistics.

**Claims And Evidence:**

No

**Claims Explanation:**

- The claims are only partially supported.  The proofs are messy and repeat arguments from prior papers verbatim without much clarity.

- The experimental results do show improvement in several tasks, in particular, cDULA and cDMALA achieve lower RMSE and higher ESS compared to DULA/DMALA on RBMs, EBMs, and binary Bayesian neural networks. However, these gains are not clearly analyzed or attributed to specific algorithmic changes, the presentation is cluttered, and the experimental setup lacks clarity and reproducibility. While the quantitative trends are promising, I feel that the evidence is not discussed in a convincing or insightful way.

**Requested Changes:**

Here, I will just summarize some key points from the **Summary of Contributions**:

- Streamline the writing **a lot** and clearly state the paper’s contribution relative to existing methods (especially Zhang et al., 2022). Do not use verbatim text from other papers and do not repeat existing results.

- Clarify why the proposed bounded projection is not a trivial extension of Zhang et al. (2022).

- Number every equation and ensure consistent notation throughout.

- Use standard probabilistic terminology (random variable, support, probability mass function, etc.).

- Justify the need for the MH correction and discuss its impact on performance and acceptance rate in high dimensions.

- Provide clearer, reproducible experimental details and consistent figure captions.

- Fix obvious mathematical and typographical errors throughout (there are many).

- Explain why the unadjusted variant (cDULA) appears to outperform the MH-corrected version (cDMALA), which is counterintuitive and suggests possible implementation or tuning issues.

- Revisit the appendix and provide a concise, structured theoretical summary of results.

Additional notes:

- The entire paper would benefit from extensive rewriting by a fluent English speaker.

- The work lacks clarity, precision, and originality in its current form.

- I feel that without substantial rewriting, this submission may not meet TMLR publication standards.

---

> ### Author Response · Authors · 2025-11-18
>
> **Q1: The Introduction (and the whole paper) can benefit from a more coherent flow that does not jump from topic to topic in every sentence. For instance, already the first paragraph is a mixture of LLM-tuned text and short sentences that do not connect well. Lots of jargon is used, which is familiar to MCMC practitioners and developers but may not be standard vernacular for ML researchers.**
>
> A1: We apologize for the lack of a detailed background on discrete sampling and MCMC-related algorithms in the previous version, which may have hindered understanding for researchers from other fields. In the revised version, we have expanded the content in greater detail, and the manuscript has been extended to over twelve pages (long submission) to better address these topics.
>
> **Q2: The mathematical formulation is inconsistent....... This sign inconsistency appears throughout the paper and undermines the connection to energy-based modeling.**
>
> A2: We appreciate the reviewers' comments, but we need to clarify that our method is based on probability distributions, which typically do not have explicit temperature parameters. More importantly, our choice to use the unnormalized distributions is in line with prior work (e.g., Grathwohl et al., 2021; Sun et al.,2023; Zhang et al., 2022). This approach allows us to express the probability distribution naturally, without unnecessary complexity. We will make this point clearer in the manuscript to avoid any confusion.
>
> [1]Grathwohl et al. "Oops I Took A Gradient: Scalable Sampling for Discrete Distributions." ICML, 2021.
>
> [2]Zhang et al. "A Langevin-like sampler for discrete distributions." ICML, 2022.
>
> [3]Sun et al. "Any-scale balanced samplers for discrete spaces." ICLR, 2023.
>
> **Q3: Similarly to the Introduction,... involves a differentiable energy function with respect to the discrete random vector which already assumes some differentiable extension but uses the same symbol.**
>
> A3: We sincerely thank the reviewer for the constructive feedback and for pointing out the typos. In the revised version, we have expanded the explanation of both the Langevin algorithm and the discrete Langevin proposal (DLP). When we describe DLP as “the counterpart of the Langevin algorithm in discrete domains,” we mean this in two senses:
>
> (1) As shown in Equation 4, when $\Theta = \mathbb{R}^d$, we have $Z_{\Theta}(\theta) = (2\pi\alpha)^{\frac{d}{2}}$, which corresponds exactly to the classical ULA (i.e., the Langevin algorithm in continuous spaces). When $\Theta$ is restricted to a discrete domain, DLP constructs an analogous proposal distribution in a similar functional form. Perhaps the phrase “Langevin-like proposal” would better capture this connection.
>
> (2) DLP is able to update all coordinates in parallel within a single iteration, where the magnitude of each update is controlled by a stepsize—again mirroring the key characteristics of continuous Langevin dynamics.
>
> **Q4: Having read Section 4, ... unclear how to sample from the distribution defined in Eq.7. The text tends to oscillate between heuristic interpretation and formalism without committing to either.**
>
> A4: We appreciate the reviewer’s thoughtful comments and are glad to clarify the novelty of our work. Our main contribution lies in introducing the continuous intermediates that enable the construction of more effective proposals, thereby accelerating the sampling process (now as shown in Eq. 6). Building upon this, we further employ a projection step to prevent large moves, whose necessity is empirically confirmed by acceptance rate analyses in Appendix C. This design choice was primarily motivated by empirical observations.
>
> We also clarify that DLP itself does not involve “relaxation” or “projection”; the reviewer may have conflated the differentiable extension of the unnormalized energy function with the extension of the sample space. Specifically, while many discrete energy functions (e.g., the Ising model) are differentiable with respect to $x$, $\log p(x)\propto x^TWx + b^Tx$, whose gradient $\nabla \log p(x) \propto 2W x + b$ is well defined even if $x \in \{0,1\}^d$, our method instead introduces continuous intermediates defined over a relaxed domain (e.g., $[0,1]^d$ for binary variables). This idea stems from the observation that such energy functions preserve informative gradient structure beyond discrete points — information that existing discrete samplers fail to exploit.
>
> To understand how to sample from such a distribution, consider a simple binary variable $\theta \in \{0,1}\$. Suppose the current state is $\theta_0=0$; using Eq. 8, we construct a discrete proposal distribution such as $[0.6,0.4]$, meaning $P(\theta_1=0|\theta_0=0)=0.6$ and $P(\theta_1=1|\theta_0=0)=0.4$. The next state is then sampled accordingly. For more complex cases, the fourth subfigure in Figure 1 (from left to right) visualizes such discrete proposal distributions, where the circle sizes represent the relative probabilities.

---

> ### Author Response · Authors · 2025-11-18
>
> **Q5: I failed to see how applying the MH correction (Section 4.3) does not inherit all limitations (e.g., curse of dimensionality) of basic MH MCMC. ... language to describe the steps (Explore -> this is rather an update step; map -> this should be relaxation + projection, etc)**
>
> A5: We agree with the reviewer that traditional MH-based MCMC methods (e.g., Gibbs, D-SVGD, R-HMC, R-MALA) often suffer from the curse of dimensionality in discrete spaces. To alleviate this issue, DLP and related methods adopt a coordinate-factorized assumption that allows all coordinates to be updated in parallel, thereby reducing the computational cost of proposal generation from $O(S^d)$ to $O(d)$. We follow the same assumption.
>
> The motivation for introducing the MH correction (Section 4.3) is to ensure that the Markov chain attains the desired stationary distribution. In Section 5.1, we prove that the chain is reversible for log-quadratic distributions; however, reversibility is not guaranteed for more general distributions. Therefore, we include the MH correction to ensure it is reversible when needed for broader applicability, like EBMs (Table 1).
>
> We also acknowledge the reviewer’s concern about inconsistent notation and terminology in Algorithm 1 (e.g., Explore -> update, map -> relaxation + projection). We have corrected these inconsistencies and clarified the terminology in the revised version.
>
> **Q6: The results do seem encouraging and the authors clearly put quite some effort into the experimental validation. However, ... Generally, I believe it may be helpful to extend the paper beyond 12 pages and delegate an entire experiment to the Appendix (e.g., 6.3, which seems to merely compare several samples qualitatively).**
>
> A6: We thank the reviewer for recognizing the effort we invested in the experimental validation. We sincerely apologize for the lack of clarity in the experimental descriptions. In the revised version, we have added detailed explanations of the experimental motivation, interpretations of the observed results, and potential directions for improvement.
>
> Figure 3 illustrates sampling from a synthetic multimodal discrete distribution. The goal was to visually demonstrate the effectiveness of continuous exploration, which is faster discovery of multiple modes and smaller total variation (TV) distance from the target distribution. The choice of 25 modes was not essential but intended to emphasize multimodality in discrete distributions. As another reviewer suggested, we will move Figure 7 to the main text for clearer presentation.
>
> In Figure 6, which reports results on the Ising model learning task, the curve of DMALA is mostly overlapped by that of cDMALA, only slightly visible above it. We apologize for the confusion this caused. In the revised version, we have redrawn the figure using lines with distinct thicknesses to improve visibility, though we note this may slightly affect the visual impression of relative performance.
>
> **Q7: The same goes for the theoretical results. .... Theorem 2 appears to simply restate Theorem 1 under weakly perturbed conditions.**
>
> A7: We thank the reviewer for carefully checking our theoretical section. We have thoroughly reviewed and corrected all typographical errors, including inconsistent dimensions in matrix–vector products. While we follow the asymptotic proof framework of Zhang et al. (2022), the adaptation to our setting is non-trivial. The introduction of continuous intermediates leads to additional terms involving $\alpha_0$ and gradient coupling in both $q$ and $Z$. Fortunately, these terms are bounded by controllable constants, allowing us to extend the theoretical guarantees while preserving convergence properties. Also, we provide a non-asymptotic proof for cDMALA, which is absent in the discrete Langevin proposal.
>
> **Q8: The connection between the main results and appendices is extremely weak...., making reproducibility impossible.**
>
> A8: We appreciate the reviewer’s suggestion and have significantly improved the organization and clarity of the Appendix. We now provide a comprehensive table summarizing all hyperparameters to facilitate reproducibility. We have also strengthened the connection between the main text and the supplementary material.
>
> In summary, we did not extensively fine-tune parameters for each experiment, except for adjusting $\alpha$ in cDULA. Motivated by its exploratory nature (Figure 2(a)), we used larger $\alpha$ values, under which cDULA demonstrated stable performance, whereas DULA tended to diverge early under the same stepsize. To prevent continuous intermediates from straying too far from their local neighborhood, we correspondingly set smaller $\alpha_0$ values in all experiments.

---

### Author Response · Authors · 2025-11-24
**Revised Manuscript**

Dear Action Editor and Reviewers,

Thank you for your constructive and insightful comments on our work. We have uploaded the revised manuscript accordingly, and all main modifications are highlighted in blue in the updated version.

Our revisions mainly focus on the following aspects:

- We have corrected all typos and issues related to formula notation throughout the manuscript to ensure accuracy and clarity.

- We have rewritten the description of our contributions to more clearly highlight the novelty and unique aspects of our method.

- We have added further discussion on the extensibility of our approach, elaborating on how the proposed framework can be applied or generalized to broader settings.

- We have provided additional clarifications for parts of the main text that were previously insufficiently explained, directly addressing the reviewers’ concerns.

- We have included in the appendix a detailed explanation of the selection of key hyperparameters in our sampler, offering clearer guidance for reproduction.

Because the length of the revised manuscript exceeds 12 pages, we have selected the ''Long submission'' format.

If there are any further comments or questions, please feel free to let us know.

Best regards,

Authors

---

### Decision · Action_Editor_weSf · 2025-12-21

**Recommendation:** Reject

**Audience:**

No

**Audience Explanation:**

The paper proposes two types of gradient-based discrete Langevin samplers: the “Discrete Unadjusted Langevin Algorithm via Continuous Intermediates” (cDULA) and the “Discrete Metropolis Adjusted Langevin Algorithm via Continuous Intermediates” (cDMALA). In contrast to previous approaches, these methods perform a continuous gradient step followed by a projection step to construct more informed proposal distributions and achieve faster convergence. An optional Metropolis–Hastings correction is introduced to ensure reversibility.

The initial reviews of this paper were mixed. While some reviewers acknowledged the hybrid approach, the theoretical analysis, and the promising empirical performance, there were also significant concerns. Although the authors were able to address some of these concerns (e.g., computational overhead, factorization & scaling, more in-depth discussion), a number of substantial weaknesses remain, including the following:

- **Presentation.** The paper’s structure follows (Zhang et al., 2022) very closely, including:

  - (1) The construction of the proposal distribution in Eq. (8) (cf. Zhang et al., 2022; Eq. (1)) and the definition of the Metropolis–Hastings correction in Eq. (11) (cf. Zhang et al., 2022; Eq. (3)), which differ only in the use of continuous intermediates - obtained via projected gradient ascent - instead of discrete states.

  - (2) The derivation of Theorems 1 and 2, which employ the same analytical framework as, and are direct extensions of, (Zhang et al., 2022; Theorems 5.1 and 5.2); even the section titles and verification experiments are identical.

  - (3) The experimental evaluations on Ising models in Section 6.2 (cf. Zhang et al., 2022; Section 7.1), RBMs in Section 6.3 (cf. Zhang et al., 2022; Section 7.2), EBMs in Section 6.4 (cf. Zhang et al., 2022; Section 7.3), and BNNs in Section 6.5 (cf. Zhang et al., 2022; Section 7.4), which reproduce descriptions in (Zhang et al., 2022) almost verbatim. While the reproduction of an established evaluation framework is generally desirable, taking sections with minimal modifications from another paper is problematic.


- **Projection Step.** The projection step is restricted to a class of spaces $K$ with special structure (e.g., hypercubes), whereas the general case does not have an efficient solution. Many discrete domains cannot be relaxed to such a tractable region $K$.

- **Convergence Analysis.** The distributional assumptions (e.g., [approximately] log-quadratic distributions in Theorems 1 & 2) and opaque dependencies (e.g., constants $c, c_1$, $c_2$ in Theorems 1 & 2) make it difficult to assess the practical relevance or scaling behaviour of these bounds in real-world scenarios. The simple experiments presented in Figure 2 (e.g., $2\times 2$ Ising model; 1D distribution) offer only limited insight in this regard.

Due to these weaknesses, the presented findings are unlikely to be of interest to TMLR's audience. Addressing these issues requires a level of revision that goes beyond what is feasible in a minor revision. I therefore cannot recommend acceptance of the manuscript in its current form and encourage the authors to address the concerns above and consider resubmission at a later date.

**Claims And Evidence:**

No

**Claims Explanation:**

The paper claims that the proposed approach enables the sampler to “better utilize the underlying geometry", but beyond reporting quantitative improvements, it provides little intuition or qualitative insight to support this claim.

From a theoretical perspective, the contribution is clearly articulated; however, the accompanying proofs are poorly organized and difficult to follow.

The experimental results are promising and indicate superior performance compared to several well-known baselines. Nevertheless, the official reviews and official recommendations raise legitimate concerns about the comparability of these results to prior work, as well as about missing details required for reproducibility; the inclusion of a hyperparameter table is a positive step in this regard.

**Resubmission Of Major Revision:**

The authors may consider submitting a major revision at a later time.